# Effects of Wind Wave Spectra, Non-Gaussianity, and Swell on the Prediction of Ocean Microwave Backscatter with Facet Two-Scale Model

**Yuqi Wu** [1], **Chenqing Fan** [2,3], **Qiushuang Yan** [1,*], **Junmin Meng** [2,3], **Tianran Song** [1] **and Jie Zhang** [1,2,3]

[1]  College of Oceanography and Space Informatics, China University of Petroleum, Qingdao 266580, China
[2]  First Institute of Oceanography, Ministry of Natural Resources, Qingdao 266061, China
[3]  Technology Innovation Center for Ocean Telemetry, Ministry of Natural Resources, Qingdao 266061, China
[*]  Correspondence: yanqiushuang@upc.edu.cn

**Abstract:** The image intensity of high-resolution synthetic aperture radar (SAR) is closely related to the facet scattering distribution. In this paper, the effects of wind wave spectra, non-Gaussianity of the sea surface, and swell on the distribution of the facet normalized radar cross section (NRCS) simulated by the facet two-scale model (TSM) are analyzed by comparing the simulated results with the Sentinel-1 SAR data, the Advanced Scatterometer (ASCAT) data, and the geophysical model function (GMF) at the wind speed range of 3–16 m/s, the wind direction range of 0°–360°, and the incidence angle range of 30°–50° under VV and HH polarizations. The results show that the Apel spectrum achieves a more consistent mean NRCS and NRCS distribution with the reference data at low incidence angles, while the composite spectra perform better at high incidence angles under VV polarization. Under HH polarization, the Apel spectrum always has a better performance. The upwind–downwind asymmetry of backscattering can be predicted well by the modified TSM, which is constructed by incorporating bispectrum correction into the conventional TSM. The distribution of the scattering simulated by the modified TSM deviates from the Gaussian distribution significantly, which is in good agreement with the Sentinel-1 data. Additionally, the introduction of swell widens the spread of the NRCS distribution, and the fluctuation range of the NRCS profile considering swell is larger than that without swell. All these changes caused by the introduction of swell make the distribution of the facet scattering more consistent with the Sentinel-1 data. Moreover, the scattering image patterns and scattering image spectrum of the Sentinel-1 data can be successfully simulated at various sea states with the consideration of swell.

**Keywords:** wind wave spectra; non-Gaussianity; swell; facet backscatter

## 1. Introduction

The theoretical investigation of microwave backscatter from the ocean surface is of great value for the geophysical interpretation of microwave remote-sensing retrievals of ocean surface winds, waves, and currents. In recent decades, research on the physical processes involved in electromagnetic–ocean-surface interactions has identified that the most important mechanism contributing to ocean backscatter at moderate incidence angles is resonant Bragg scattering. Many physical models have been developed to describe Bragg scattering, such as the small perturbation method [1], two-scale model (TSM) [2,3], small-slope approximation [4], and integral equation method [5]. However, most related studies focused on the average backscatter from a large-scale sea surface on the order of tens of kilometers and ignored fluctuations corresponding to local spatiality characteristics. Such local information is valuable and desirable in the simulation of intensity patterns acquired by a synthetic aperture radar with a high resolution.

Considering this demand, research has focused on facet-based methods that divide the ocean surface into facets and obtain the scattering contribution from each individual

facet. For example, Franceschetti et al. [6–8] presented a facet backscatter model in which the Kirchhoff solution of the physical optics approximation was used to compute the facet scattering, and then a shortwave modification of the facet was used to revise the phase factor that accounts for the microstructure (ripples). Martino et al. [9] used the small perturbation method to predict the scattering of a facet Gaussian sea surface that is isotropic. Zhang et al. [10] and Li et al. [11] built a more exact simulation of the scattering phase distribution and provided a modified TSM, called the capillary wave modification facet scattering model, to evaluate the scattering from facet non-Gaussian sea surfaces, either in the monostatic or the bistatic cases. In [12], Su et al. calculated the scattering of a nonlinear sea surface based on the capillary wave modification facet scattering model. In [13,14], Chen et al. combined the Kirchhoff approximation with the small perturbation method to obtain the bistatic normalized radar cross section (NRCS) of a Gaussian sea surface.

Most previous studies of facet scattering were carried out based on the JONSWAP spectrum [15] and the Elfouhaily spectrum [16]. The JONSWAP spectrum, developed by Hasselmann et al. in 1973, is a gravity wave spectrum [15]. The Elfouhaily spectrum is a unified spectrum that combines the JONSWAP wave spectrum and the short-wave spectrum derived from wave tank data [17]. However, many wind wave spectra have been developed to date. Different spectra have different performances in NRCS prediction under various incidence angles and wind speeds. Therefore, it is necessary to analyze the influence of wind wave spectra on the prediction of facet scattering. The influence of the non-Gaussianity on facet scattering simulation also needs to be further discussed at different incidence angles, wind speeds, and wind directions. Additionally, the swell, which can not be ignored in the actual ocean environment, is a significant factor as well. Swell can change the height fluctuation and slope distribution of the sea surface, and thus it can cause changes in scattering patterns. However, most studies focused on the influence of swell on the mean NRCS prediction and ignored the impact of swell on the simulation of scattering distribution. So, it is necessary to discuss the effect of swell on facet scattering.

In this study, we analyze the effects of wind wave spectra, non-Gaussianity, and swell on the distribution of facet scattering predicted with TSM by comparing simulated results with the Sentinel-1 data, the Advanced Scatterometer (ASCAT) data, and the geophysical model function (GMF) at the incidence angles of 30°–50° over a wind speed range of 3–16 m/s and a wind direction range of 0°–360° under C-band VV and HH polarizations. We used the TSM to calculate the facet radar backscatter because the TSM has been the most popular analytical scattering model since it was first presented in the 1960s owing to its reliable precision and high efficiency. The remainder of this paper is organized as follows: Section 2 briefly introduces the reference data, the method used to construct the sea surface, and the model to calculate the facet scattering. In Section 3, we analyze the influence of wind wave spectra, non-Gaussianity, and swell on the performance of simulated facet scattering by comparing the simulated results with the reference data. The discussion and conclusions are presented in Sections 4 and 5, respectively.

## 2. Data and Methods

### 2.1. Reference Data

2.1.1. Sentinel-1 SAR Data

The Sentinel-1 satellite was designed by the European Space Agency (ESA). It performs C-band imaging in the four exclusive modes of strip map mode (SM), interferometric wide-swath mode (IW), extra wide-swath mode (EW), and wave mode (WV). The Sentinel-1 SAR data in the IW and EW modes have a large swath width of hundreds of kilometers and can provide data for a wide range of incidence angles. In this paper, we collected about three thousand Sentinel-1 ground range detected (GRD) high-resolution images in the IW (pixel spacing is 10 m × 10 m) and EW modes (pixel spacing is 25 m × 25 m) under dual polarization (HH + HV, VV + VH) with incidence angles ranging from 32° to 45°. We collected data in the Southern Ocean for the past four years and data near Australia for the past two years.

Since a Sentinel-1 image is too large, we first cropped the entire Sentinel-1 image and obtained a 10 km × 10 km sub-image every 50 km × 50 km. Then, in order to obtain the wind and swell information corresponding to the Sentinel-1 data, we matched the 10 km × 10 km sub-images with the time/space interpolated sea surface winds and ocean waves obtained from the fifth-generation reanalysis (ERA5). The ERA5 is produced by ECMWF and provides a detailed record of the global atmosphere, land surface, and ocean waves from 1950 onwards [18]. Finally, we obtained about 59,788 matching pairs.

### 2.1.2. ASCAT Data

The Advanced Scatterometer (ASCAT) is carried on the meteorological operational (MetOp) platforms that constitute the space segment of the European Organisation for the Exploitation of Meteorological Satellites (EUMETSAT) Polar System (EPS). It is a real aperture C-band VV-polarized radar with incidence angles ranging from 25° to 65°. It is well calibrated using transponders with an absolute calibration precision better than 0.25 dB [19]. The ASCAT level-1B 12.5 km $\sigma^0$ product produced by EUMETSAT from January 2014 to December 2016 was used. The ASCAT $\sigma^0$ measurements influenced by non-wave phenomena were rejected based on the following principles: (1) the default filling data and the abnormal data flagged as bad-quality were excluded; (2) the measurements contaminated by rainfall were eliminated (the rainfall information was provided by ECMWF); and (3) the measurements contaminated by land/island were eliminated.

Then, the $\sigma^0$ measurements of ASCAT were collocated with the buoy observations of sea surface winds obtained from the National Data Buoy Center (NDBC) with the following criteria: time separation within 30 min and spatial separation of less than 25 km. If one buoy datum matches several ASCAT $\sigma^0$ data, we used the one with the closest distance. Finally, 3,502,466 matching pairs were obtained. The buoys were all located in waters more than 50 km away from land and 150 m deep. The buoy wind speeds sampled at 2–5 m heights above the sea surface were converted to 10 m neutral wind speeds based on the logarithmic marine boundary layer assumption proposed by Liu and Tang [20] after eliminating low-quality observations.

### 2.1.3. Geophysical Model Function

The GMF, CMOD5.n, provides the C-band VV NRCS as a function of the incidence angle, wind direction (relative to the radar look direction), and 10 m neutral wind speed [21]. It takes the following general form:

$$\sigma^0(\theta, U_{10}, \varphi) = B_0(\theta, U_{10})(1 + B_1(\theta, U_{10})\cos\varphi + B_2(\theta, U_{10})\cos 2\varphi)^{1.6} \tag{1}$$

where $\sigma^0$ is the NRCS in natural units (not in decibels), $\theta$ is the incidence angle, $U_{10}$ is the 10 m neutral wind speed, and $\varphi$ is the azimuthal wind direction angle. $B_0$, $B_1$, and $B_2$ are functions of $\theta$ and $U_{10}$ and were obtained by fitting (1) to the European Remote Sensing Satellite (ERS) scatterometer data. The mean absolute difference between CMOD5.n and the ASCAT measurements was about 0.3 dB, and the root-mean-square difference was less than 2 dB [22].

When such a GMF is applied to HH-polarized scattering, polarization ratio (PR) models have to be used for converting VV- into HH-NRCS. A prevalent C-band PR model provided by Thompson et al. [23,24] was used in this paper:

$$\text{PR} = \frac{\left(1 + 2\tan^2\theta\right)^2}{\left(1 + \alpha\tan^2\theta\right)^2} \tag{2}$$

where $\alpha = 0.6$.

*2.2. Ocean Wave Spectrum*

The construction of a large-scale sea surface is a significant step in the facet scattering calculation, which highly depends on the wind wave spectrum. Therefore, in order to study the facet scattering from the sea surface, it is necessary to acquire knowledge of the sea spectrum.

In recent decades, many full wavenumber spectra have been proposed, of which the Donelan–Banner–Plant spectrum (D spectrum) [25], Apel spectrum (A spectrum) [26], Elfouhaily spectrum (E spectrum) [16], and the newest Hwang spectrum (H18 spectrum) [27] were used in this paper. The D spectrum considers the synthesis of DHH [28], Banner [29], and Plant [30] into a single spectral form. The A spectrum was derived from in situ measurements in the gravity wave region and from wave tank results in the gravity-capillary wave region. It addresses the consistency with observational NRCS data. The E spectrum was developed based on the unification of optical, in situ, and wave tank data, and it emphasizes the consistency with the measured mean square slopes in [31]. The H18 spectrum is a combination of the H15 spectrum [32] and the G spectrum [27]. In the high-frequency region (wavenumber $\geq 4\,\mathrm{rad/m}$), the H18 spectrum uses the H15. In the low-frequency region (wavenumber $\leq 1\,\mathrm{rad/m}$), it uses the variable spectral slope model, the G spectrum. Additionally, the spectrum in the region of $1 \leq$ wavenumber $\leq 4\,\mathrm{rad/m}$ is linearly interpolated. The details of the four spectral models are presented in [16,25–27] and will not be repeated here.

Figure 1 gives the variation of D, A, E, and H18 omnidirectional spectra $S(K)$ and curvature spectra $B(K) = K^3 S(K)$ with wavenumber $k$ at wind speeds of 5 m/s, 10 m/s, and 15 m/s. It can be seen from Figure 1a–c that the omnidirectional spectra exhibit unimodal distribution features, with peaks occurring in the low wavenumber region. As wind speed increases, the peaks of spectra shift toward the low wavenumber region, and the power of each wind wave spectral component also increases in level. Figure 1e,f give the omnidirectional spectra $S(K)$ that account for Bragg scattering for incidence angles between about $30°$ and $60°$ in the C band. From Figure 1e,f, we can know that the A spectrum's density is always the largest and the D spectrum's density is the smallest, except that the density of the A spectrum is slightly smaller than that of the E spectrum at the wind speed of 5 m/s when the wavenumber is greater than 140 rad/s. The curvature spectral densities of wind wave spectra in Figure 1h–j show a decreasing wind speed sensitivity as wind speed increases. In addition, there is a spectral peak in the capillary wave region, followed by a rapid drop-off. For a given wind speed of 15 m/s, the peak is at about 200 rad/s for the D spectrum, 700 rad/s for the A spectrum, and about 400 rad/s for the E and H18 spectra.

The spectra mentioned above were used to describe wind waves, and the influence of the swell can be described by the swell spectrum, which can be expressed as follows:

$$S_{\text{swell}}(k_x, k_y) = \frac{\langle h^2 \rangle}{2\pi\sigma_{Kx}\sigma_{Ky}} \exp\left\{ -\frac{1}{2}\left[ \left(\frac{k_x - k_{xm}}{\sigma_{Kx}}\right)^2 + \left(\frac{k_y - k_{ym}}{\sigma_{Ky}}\right)^2 \right] \right\} \tag{3}$$

where $\langle h^2 \rangle$ is the root-mean-square height of the simulated swell, and its root is 1/4 of the significant wave height of swell $H_s$. $x$ is the horizontal antenna look direction (ground range direction), and $y$ is the perpendicular horizontal direction (azimuth direction). $k_x$ and $k_y$ represent the wavenumber components of the wavenumber $k$ on the range and azimuth coordinates, respectively. $k_{xm}$ and $k_{ym}$ denote the wavenumber at the peak of the spectrum in the range and azimuth axis direction, respectively. The spectral widths in two coordinate directions are represented by $\sigma_{Kx}$ and $\sigma_{Ky}$, and in this paper, $\sigma_{Kx} = \sigma_{Ky} = 0.0025\,\mathrm{m}^{-1}$. Figure 2 shows the swell spectrum with different significant wave heights of swell $H_s$ and swell wavelengths $\lambda_c$. Figure 2a,b show that for a certain $H_s$, the wavelength of swell has little effect on the intensity of the swell spectrum. From Figure 2b,c, we can see that $H_s$ affects the spectral intensity, which will become larger as $H_s$ increases.

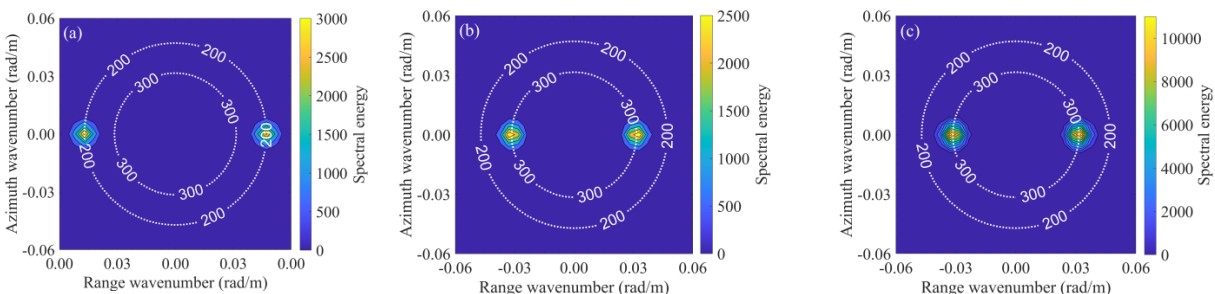

**Figure 1.** The variation of D, A, E, and H18 omnidirectional spectra $S(k)$ and curvature spectra $B(k)$ with wavenumber k at wind speeds of 5 m/s, 10 m/s, and 15 m/s. (**a**–**c**) Omnidirectional spectra vary with wavenumber k, (**d**–**f**) omnidirectional spectra that account for Bragg scattering for incidence angles between about 30° and 60° in the C band, (**g**–**i**) curvature spectra vary with wavenumber k. Green curve, D spectrum; blue curve, A spectrum; red curve, E spectrum; magenta curve, H18 spectrum.

**Figure 2.** The swell spectrum with different significant wave heights and dominant wave wavelengths. (**a**) The significant wave height of swell = 2 m, swell wavelengths = 200 m; (**b**) the significant wave height of swell = 2 m, swell wavelengths = 300 m; (**c**) the significant wave height of swell = 4 m, swell wavelengths = 300 m.

### 2.3. Construction of the Sea Surface

The Gaussian sea surface is envisaged as a series of superpositions of cosine waves with different wavelengths, amplitudes, and phases by the linear superposition method [33,34]. The surface elevation can be written as follows:

$$z_1(x_m, y_n, t) = \sum_{x=1}^{M} \sum_{y=1}^{N} \sqrt{2S(k,\varphi)dk} \cos\left(k_x x_m + k_y y_n - \omega_{k_x,k_y} t + \varepsilon_{k_x,k_y}\right) \tag{4}$$

where $M$ and $N$ are the numbers of facets in ground range and azimuth direction, respectively. $S(\cdot)$ is the ocean wave spectrum, in which the spreading function of the cosine form [16] is used. $\omega_{k_x,k_y}$ is the angular frequency, and $\varepsilon_{k_x,k_y}$ is uniformly distributed over $(0, 2\pi)$. The coordinate system can be expressed as follows:

$$\begin{aligned} x_m &= \Delta x(m - M/2)|_{m=1,2,\dots,M} \\ y_n &= \Delta y(n - N/2)|_{n=1,2,\dots,N} \end{aligned} \tag{5}$$

in which $\Delta x$ and $\Delta y$ are the size of the facet in ground range and azimuth direction.

Nevertheless, the actual ocean environment is extremely complicated and usually has characteristics of non-Gaussianity. In this paper, we used the Tayfun model [35] to construct a non-Gaussian large-scale sea surface. The Tayfun model can be expressed as follows:

$$z(x) = z_1 + \frac{1}{2}\widetilde{k}\left[\left(z_l^2 - \widetilde{z}^2\right) + 2z_l\widetilde{z}\right] \tag{6}$$

where $\widetilde{k}$ is the mean wavenumber, $\widetilde{k} = m_1/m_0$, $m_j = \int k^j S(k)dk$, and

$$\widetilde{z}(x_m, y_n, t) = \sum_{x=1}^{M} \sum_{y=1}^{N} \sqrt{2S(k,\varphi)dk} \sin\left(k_x x_m + k_y y_n - \omega_{k_x,k_y} t + \varepsilon_{k_x,k_y}\right) \tag{7}$$

### 2.4. Facet Scattering Model

In this paper, the TSM was used to calculate the facet scattering of the sea surface. Under the assumption that the ocean surface consists of small-scale roughness superimposed on a large-scale surface, TSM combines the Bragg scattering from the small-scale roughness, whose wavelengths are in the order of the electromagnetic wave, and the local tilting and hydrodynamic modulations due to the longer underlying waves. It can be expressed as follows:

$$\sigma_{pp}^s(\theta') = 16\pi k_i^4 \cos^4\theta' G_{pp} S(K_B, \varphi) \tag{8}$$

where $\theta'$ is the local incidence angle of each tilting facet related to the inclination of the large-scale sea surface. The subscript $pp$ indicates the VV or HH polarization state, $k_i$ is the radar wavenumber, and $G_{pp}$ is the VV or HH scattering coefficient. $K_B = 2k_i\sin\theta$ is the Bragg resonance wavenumber, and $\theta$ is the incidence angle.

However, the conventional TSM cannot describe the upwind–downwind asymmetry of microwave backscatter from the ocean surface. Because the simulated NRCSs based on conventional TSM are 180 degrees symmetric in the azimuth direction, we developed a modified TSM by adding a backscattering correction term proportional to the bispectrum function into the conventional model. The NRCS for each individual facet of the surface can be given as follows:

$$\sigma_{pp}^0(\theta') = \sigma_{pp}^s(\theta') + \sigma_{pp}^c(\theta') \tag{9}$$

where $\sigma^c$ is the complementary term to consider the third-order statistics of the sea surface. According to [36], $\sigma^c$ can be expressed as follows:

$$\sigma_{pp}^c(\theta') = -k_i^5 \cos^3\theta' B_a(K_B, \varphi)\left[4|f_{pp}|^2 + 1.5\mathrm{Re}\left[f_{pp} \cdot F_{pp}\right] + 0.125|F_{pp}|^2\right] \tag{10}$$

where $f_{pp}$ and $F_{pp}$ are the polarization-dependent coefficients, and $B_a$ is the imaginary part of the bispectrum. $B_a$ can be written as follows:

$$B_a(K_B, \varphi) = -\frac{K_B s_0^6 (6 - K_B^2 s_0^2 \cos^2 \varphi) \cos \varphi}{16} \exp\left(-\frac{K_B^2 s_0^2}{4}\right) \tag{11}$$

where $s_0$ is the skewness parameter, which can be expressed as follows [37]:

$$s_0 = \zeta \xi \frac{\sigma_R}{(U_{12.5} - A/B)^{1/3} U_{12.5}^{1/2}} \tag{12}$$

$$\xi = \frac{(6/B)^{1/3}}{\sqrt{0.5C}} \tag{13}$$

where $A = 5.0 \times 10^{-2}$, $B = 42 \times 10^{-3}$, and $C = 5.1 \times 10^{-3}$. $A$ and $B$ are the coefficients in the relation between skewness coefficients and wind speed proposed by Cox and Munk [32,38]. $C$ is the coefficient in the expression of the total slope variance as a function of the wind speed. $U_{12.5}$ denotes the wind speed at a height of 12.5 m above the sea surface, and $\sigma_R^2$ is the vertical variance of a small-scale wave, which can then be evaluated from the following:

$$\delta = k_i \sigma_R \tag{14}$$

$$\delta = 0.205 \log_{10} u_* - 0.0125 \tag{15}$$

where $u_*$ is the wind friction velocity. $\zeta$ is an undetermined parameter and can be estimated by minimizing the difference between the upwind–downwind asymmetry predicted by the modified TSM and that provided by CMOD5.n. $\zeta$ is mainly related to the wind speed, and the dependence of $\zeta$ on the incidence angle is small [36]. Additionally, the estimated $\zeta$ values under different spectra are almost the same, with the difference being within 0.01 dB. Figure 3 shows the variations of $\zeta$ with wind speed based on the A spectrum and the cosine spreading function under VV and HH polarizations at an incidence angle of 32°. We can see from Figure 3 that for a given incidence angle, $\zeta$ is a nonlinear monotonically increasing function of wind speed, and the increasing rate speeds up gradually. The value of $\zeta$ under HH polarization is smaller than that under VV polarization, and the gap increases as the wind speed increases.

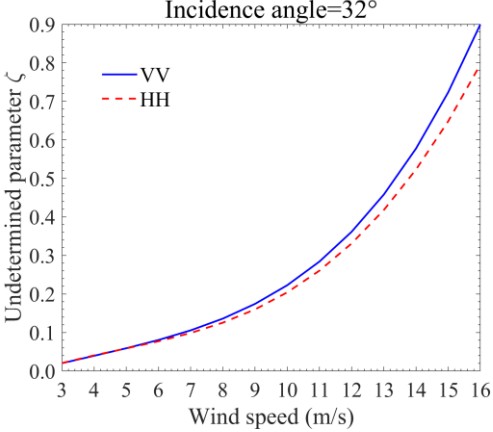

**Figure 3.** The variations of $\zeta$ with wind speed under VV and HH polarizations at the incidence angle of 32°.

## 3. Results

The effects of the wind wave spectra, non-Gaussianity of the sea surface, and swell on the backscatter prediction with the facet TSM were analyzed by comparing the simulated

NRCS with reference data over the wind speed range of 3–16 m/s, wind direction range of 0–360°, and incidence angles of 30°–50°.

### 3.1. Effect of Wind Wave Spectra on NRCS Simulation

The wind wave spectrum describes the quasi-periodic nature of ocean surface oscillations. It plays an indispensable role in the study of microwave electromagnetic scattering from the sea surface [39–42]. Therefore, the performance of different wind wave spectra in scattering prediction should be analyzed by comparing the scattering simulation obtained using the conventional TSM based on four spectra with the reference data at the wind speeds of 3–16 m/s under moderate incidence angles for VV and HH polarizations. We simulated the scattering from a Gaussian sea surface based on the D, A, E, and H18 wave spectra for a sea surface size of 2 km × 2 km and a facet size of 10 m × 10 m. Furthermore, the Sentinel-1 data, ASCAT measurements, and NRCSs calculated by the empirical model were used as the reference data. The ASCAT NRCS was averaged within ±0.5 m/s wind speed, ±6° wind direction, and ±0.5° incidence angle bins, and the Sentinel-1 NRCS was averaged within ±0.5 m/s wind speed, ±6° wind direction, and ±0.5° incidence angle bins.

The NRCS predictions of the conventional TSM from a Gaussian sea surface based on different wind wave spectra and the reference data as a function of incidence angle in upwind direction under wind speeds of 5, 10, and 15 m/s are illustrated in Figure 4. This figure shows that the NRCSs computed based on the A spectrum are the highest in level, whereas those calculated using the D spectrum are the smallest. This phenomenon can be explained by the behaviors of these spectra in the Bragg resonance wavenumber region shown in Figure 1d–f, in which the density of the A spectrum is always the largest and that of the D spectrum is the smallest in general. Figure 4 also shows that the NRCSs based on the D spectrum always underestimate backscatter under various wind speeds at the incidence angles of 30°–60° compared with the reference data.

At a low wind speed (5 m/s) and under VV polarization, the A spectrum overestimates the scattering. As the incidence angle increases, the error becomes larger and even reaches 3 dB. The E spectrum underestimates the NRCS at small incidence angles (>35°) and overestimates the scattering values at large incidence angles (<35°). The variation trend of the reference data with the incidence angle is consistent with that of the H18 spectrum, but the H18 spectrum underestimates the scattering by approximately 2 dB at the incidence angles of 30°–60°. Under HH polarization, all four spectra underestimate the NRCS compared with the Sentinel-1 data. This phenomenon might be caused by wave breaking, which greatly affects the scattering under HH polarization and cannot be described by the TSM. In this paper, we paid more attention to the Bragg scattering at moderate incidence angles, and the non-Bragg scattering caused by wave breaking was not taken into consideration. Moreover, the A spectrum performs better overall than other spectra in NRCS simulations under HH polarization because the overestimation of the A spectrum offsets part of the underestimation of TSM. Furthermore, all four spectra initially overestimate and then underestimate the backscatter in comparison with the empirical model, CMOD5.n + PR. At moderate wind speed (10 m/s) under VV polarization, the reference data are consistent with the predictions based on the A spectrum at small angles (<36°) but close to the predictions of the E spectrum at incidence angles of >44°. For incidence angles of 36°–44°, the reference data are between the results of the A spectrum and the E spectrum. The H18 spectrum is more consistent with the reference data as a function of the incidence angle. Under HH polarization, the performance of four spectra in predicting scattering is similar to that at a low wind speed (5 m/s). That is, the A spectrum has a smaller underestimation than other three spectra in comparison with the Sentinel-1 data. At high wind speed (15 m/s), the A spectrum performs well at incidence angles of 30°–60° under VV and HH polarizations, whereas the H18 spectrum can better describe the variation trend of the scattering with the incidence angle.

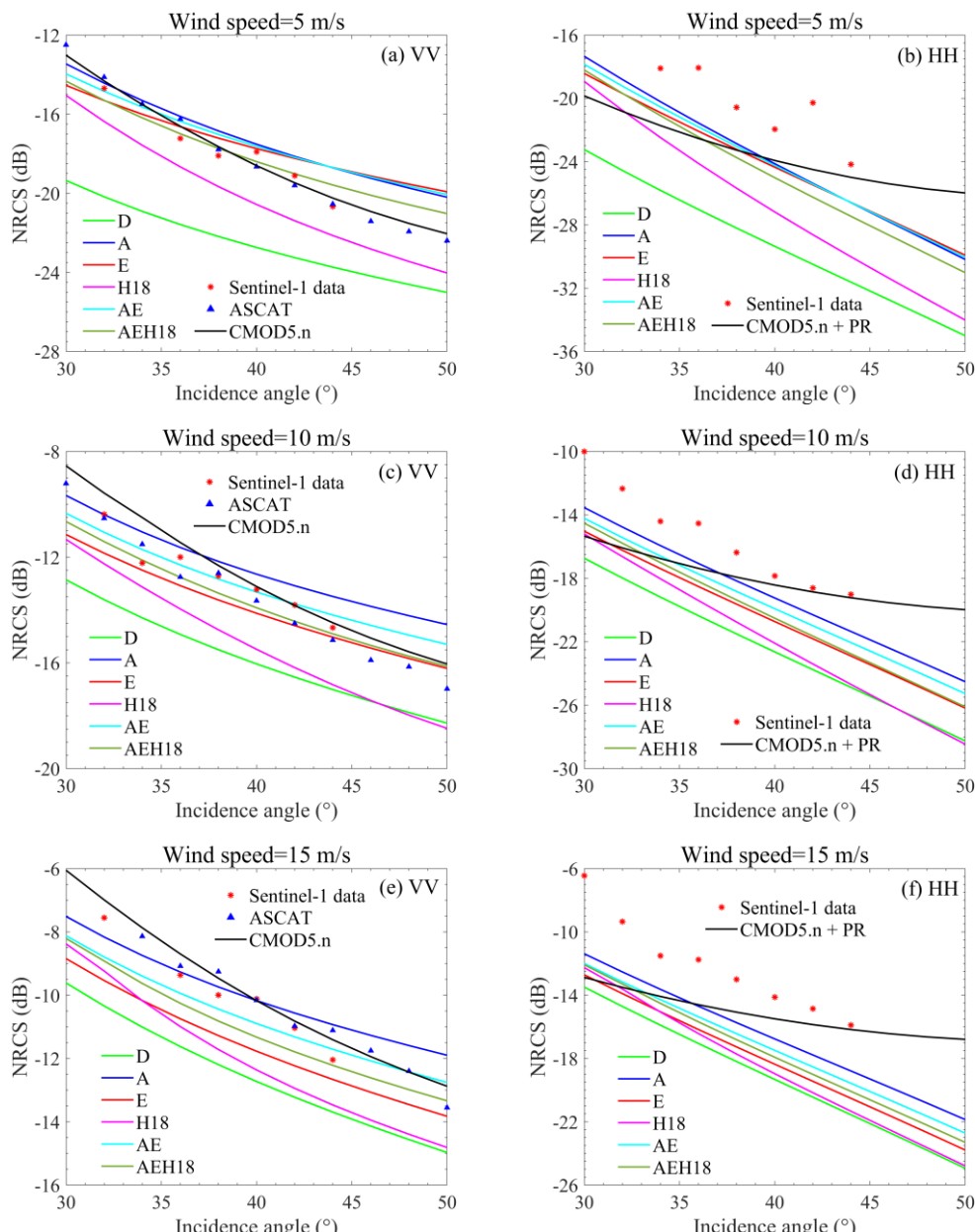

**Figure 4.** Predictions of the conventional TSM based on the four wind wave spectra of D, A, E, and H18 and two composite spectra of AE and AEH18 and their comparison with the Sentinel-1 data, ASCAT data, and empirical model as functions of the incidence angle at wind speeds of (**a**,**b**) 5 m/s, (**c**,**d**) 10 m/s, and (**e**,**f**) 15 m/s in the upwind direction under VV and HH polarizations. Green curve, D spectrum; blue curve, A spectrum; red curve, E spectrum; magenta curve, H18 spectrum; cyan curve, AE composite spectrum; dark green curve, AEH18 composite spectrum; star point, Sentinel-1 data; triangle point, ASCAT data; dark curve, empirical model.

The above analysis reveals that the satisfactory sea surface spectrum for NRCS simulation at different wind speeds and incidence angles varies, and we conjecture that a combination of these spectra might have better applicability. Therefore, we combined the A spectrum, E spectrum, and H18 spectrum, which have better performance, to obtain composite spectra. Two composite wave spectra were obtained through averaging: the average of the A and E spectra (hereafter, "AE") and the average of the A, E, and H18 spectra (hereafter, "AEH18"). Figure 4 shows that, under HH polarization, the combined spectra do not perform better than the single wind wave spectrum under different wind

speeds at all incidence angles, and the A spectrum retains the best performance in NRCS simulation. Under VV polarization, the results are similar to those described above, except that the AEH18 composite spectrum has a better applicability for incidence angles larger than 35° under low and medium wind speeds, and the AE composite spectrum yields excellent results at high incidence angles (>44°) and high wind speed (15 m/s).

Figure 5 shows the prediction of the conventional TSM using the four wind wave spectra and two composite spectra and their comparison with the reference data as a function of wind speed at the incidence angles of 32° and 45° in the upwind direction. The NRCS predictions based on the A spectrum have the largest level, whereas the results of the D spectrum are the smallest, similar to the conclusions derived from Figure 4. At a low incidence angle (32°) under VV polarization, the AEH18 combined spectrum works best when the wind speed is <5 m/s, while the A spectrum is in good agreement with the reference data at medium and high wind speeds (>5 m/s). For HH polarization, the error between the A spectrum and the reference data is the smallest, but it becomes larger in level as the incidence angle increases, even reaching 2 dB at a high wind speed (e.g., 16 m/s). The trend of the reference data with wind speed is most consistent with the H18 spectrum. At a high incidence angle (45°), the NRCS is well predicted by the AEH18 composite spectrum for the wind speed of <13 m/s and is consistent with that of the AE composite spectrum for the wind speed of >13 m/s. Under HH polarization, the A spectrum achieves a good agreement with the Sentinel-1 data at wind speeds of 3–16 m/s. The prediction by the empirical model is consistent with that of the AEH18 spectrum at low wind speeds (<7 m/s) but close to that of the A spectrum at high wind speeds (>12 m/s). When the wind speed is in the range of 7–12 m/s, the AE spectrum is in better agreement with the reference data. The above analysis reveals that the composite spectra have slightly better performance in NRCS prediction at high incidence angles in comparison with the four individual spectra. Generally, backscatter can be predicted accurately by the A spectrum, while the H18 spectrum effectively describes the trends of scattering with incidence angle and wind speed but with poor accuracy.

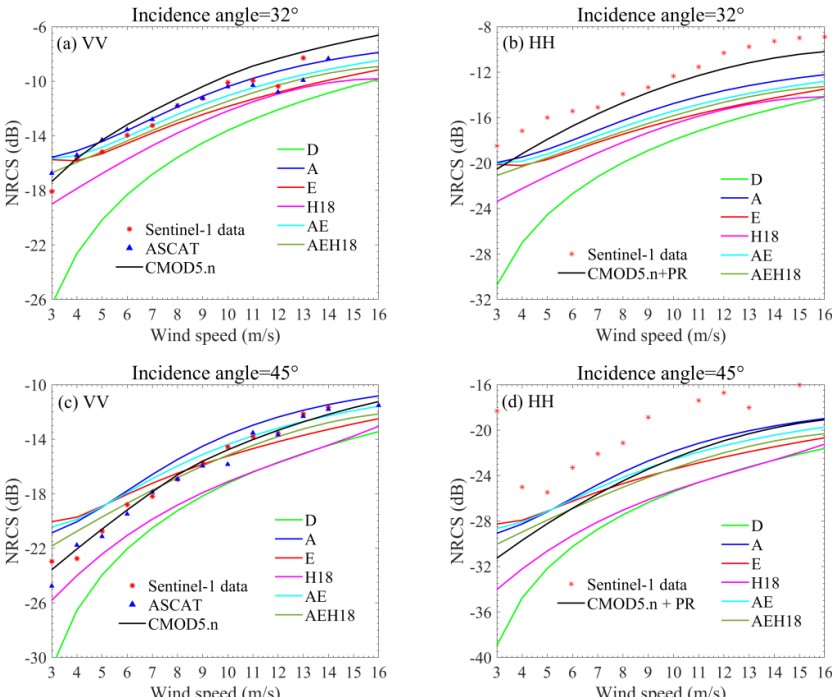

**Figure 5.** Predictions of the conventional TSM based on the four wind wave spectra and two composite spectra and their comparison with the Sentinel-1 data, ASCAT data, and empirical model as functions of the wind speed at the incidence angles of (**a,b**) 32° and (**c,d**) 45° in the upwind direction under VV and HH polarizations. Curves and symbols are the same as those in Figure 4.

Figure 6 shows examples of probability density functions (pdfs) of scattering and scattering profiles predicted by the conventional TSM with the input of the six wave spectra, and their comparison with Sentinel-1 data, for the wind speed of 10 m/s and incidence angles of 32° and 45° in the upwind direction. Considering that the sea surface resolution affects the scattering distribution (a detailed analysis is presented in Section 4), we selected the Sentinel-1 data in the IW mode, which have the same facet size as the simulation, as the reference data. Furthermore, we excluded the Sentinel-1 data including swell based on the method in [43] to eliminate the influence of swell on the scattering. The scattering pdfs shown in Figure 6a,b were obtained from multiple images, and Figure 6c,d show the scattering profile of a single image. The ranges of the scattering distributions of the Sentinel-1 data are larger than those simulated by the conventional TSM based on the four individual wind wave spectra and two composite wind spectra (Figure 6a,b). Furthermore, the peak values of the pdfs for the Sentinel-1 NRCS are always smaller than those simulated with different inputs of wave spectra. The NRCS corresponding to the peak of the scattering pdf of the Sentinel-1 data is well predicted by the A spectrum at the wind speed of 10 m/s and incidence angle of 32° and is consistent with the two composite wind spectra at the wind speed of 10 m/s and incidence angle of 45° (Figure 6a,b). In Figure 6c,d, we can see that the fluctuation range of the Sentinel-1 scattering intensity is in high agreement with the simulated results. Furthermore, the D spectrum obviously underestimates the NRCS of Sentinel-1 at the incidence angles of 32° and 45° when the wind speed is 10 m/s.

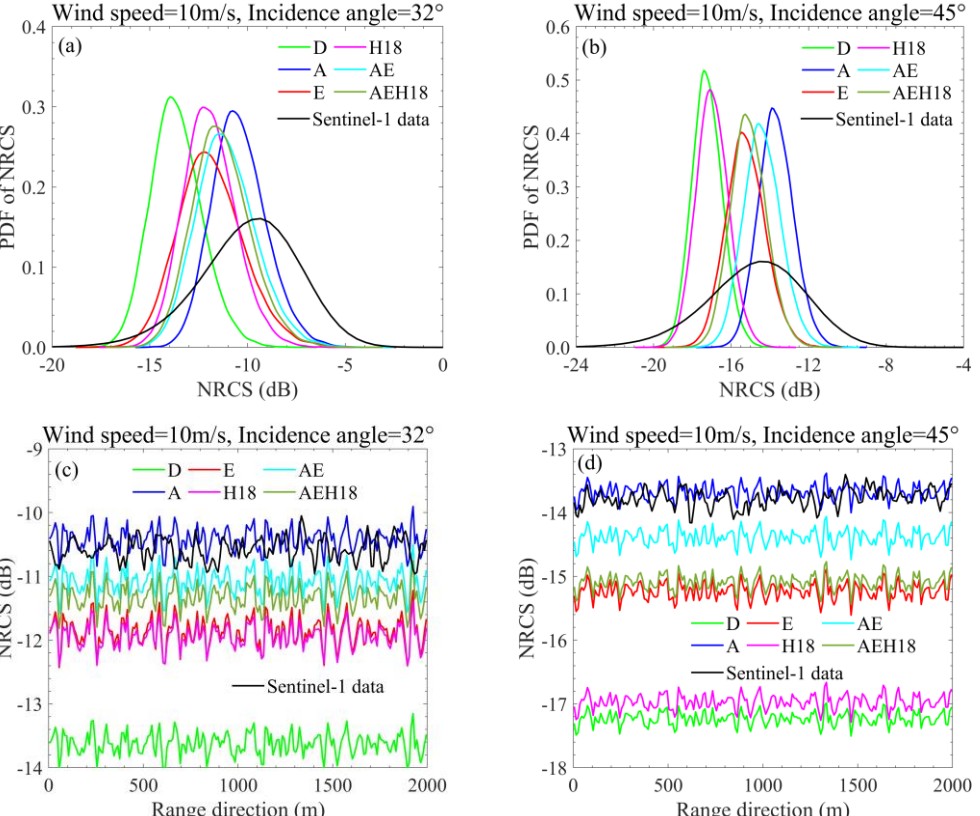

**Figure 6.** Examples of probability density functions of scattering and scattering profile predicted by the conventional TSM based on six wave spectra and their comparison with the Sentinel-1 data for the wind speed of 10 m/s at the incidence angle of 32° and 45° in the upwind direction. Curves for the different wind wave spectra are the same as those in Figure 4, and dark curves indicate Sentinel-1 data. (**a**) Pdfs for NRCSs at a 10 m/s wind speed and 32° incidence angle; (**b**) pdfs for NRCSs at a 10 m/s wind speed and 45° incidence angle; (**c**) profiles of NRCSs at a 10 m/s wind speed and 32° incidence angle; (**d**) profiles of NRCSs at a 10 m/s wind speed and 45° incidence angle.

Overall, the comprehensive analysis of the performance of NRCS simulation using the four individual wind wave spectra and two composite wind spectra reveals that the A spectrum has a better applicability for NRCS simulation at different incidence angles and wind speeds under VV and HH polarizations based on TSM, except that the combined spectra perform slightly better at high incidence angles. This conclusion is similar to that found in [44], which mentioned that the A spectrum that was developed for the EM scattering model has a high accuracy in the full wavenumber and wide wind speed range among the numerous spectra. However, in [39], the A + H18 composite wind wave spectrum has a better applicability than single wave spectra, such as the A spectrum and E spectrum. This might be because the influences of the spectra on the NRCS prediction were analyzed based on the advanced integral equation model in [39]. In the subsequent experiments, we used the A spectrum and cosine spreading function for the sea surface construction and scattering calculation.

### 3.2. Effect of Non-Gaussianity on NRCS Simulation

Previous studies focused on the scattering of a Gaussian sea surface. However, the natural sea surface is non-Gaussian, which means that it is necessary to consider the influence of non-Gaussianity on the prediction of the mean scattering and the scattering distribution. In this section, we evaluate the effect of the non-Gaussianity on the prediction of the sea surface distribution, average scattering, upwind–downwind asymmetry of backscatter, and scattering distribution at the incidence angles of 30°–50° and the wind speed range of 3–16 m/s, with the wind direction range of 0°–360° under VV and HH polarizations. The influence of non-Gaussian properties on scattering is introduced from two aspects. The first is the introduction of a non-Gaussian large-scale sea surface using the Tayfun model. The second is the introduction of the long-wave hydrodynamic modulation on the shortwave spectrum, which can be described using the modified TSM by adding a correction term associated with the bispectrum function into the conventional TSM.

Figure 7 shows the comparison of the Gaussian and Tayfun sea surfaces at the wind speed of 10 m/s and incidence angle of 32° in the upwind direction. Figure 7a shows that the distribution of the Tayfun sea surface elevation deviates from that of the Gaussian surface owing to nonlinear effects, and the mean value shifts to a negative value. Figure 7b shows that the Tayfun sea surface has sharper crests and flatter troughs in comparison with the Gaussian sea surface. Moreover, Figure 7c shows that the non-Gaussian sea surface has higher probability near small slopes in comparison with the Gaussian surface, and there is no significant difference between the pdfs of the Gaussian and non-Gaussian sea surface slopes near the large slopes in the linear scale. All these results are similar to the performance of the CWM reported in [45]. The repeated experiment at high wind speed (i.e., 15 m/s) shows that the Tayfun ocean surface has the obvious characteristics of sharp peaks and flat troughs with a slight horizontal offset.

The simulated average NRCSs based on the conventional TSM + Gaussian sea surface, conventional TSM + Tayfun sea surface, and modified TSM + Tayfun sea surface and their comparison with the reference data at the wind speed of 10 m/s as functions of the incidence angle in the upwind direction under VV and HH polarizations are shown in Figure 8a,b, and those in the downwind direction under VV and HH polarizations are shown in Figure 8c,d. The simulated average NRCSs and their comparison with the reference data at the incidence angle of 32° as a function of wind speed in the upwind direction under VV and HH polarizations are shown in Figure 9a,b, and those in the downwind direction under VV and HH polarizations are shown in Figure 9c,d. Figures 8 and 9 show that there is little difference between the mean NRCSs from the Gaussian and Tayfun sea surfaces based on the conventional TSM, which indicates that the introduction of non-Gaussian large-scale waves based on the Tayfun model has little effect on the prediction of the mean NRCS. In comparison, the simulated scattering based on the modified TSM is larger in the upwind direction and smaller in the downwind direction than the NRCS calculated using the conventional TSM owing to the introduction of bispectrum correction

in the modified model. Moreover, the level of the correction introduced by the modified model changes little with an increase in wind speed for a given incidence angle, and it changes little with incidence angle for a certain wind speed. The absolute value of the bispectrum correction in the upwind direction is slightly larger than that in the downwind direction at a certain incidence angle and wind speed. Additionally, the NRCS from the non-Gaussian surface predicted by the modified TSM is in good agreement with that of the Sentinel-1 data as a function of incidence angle and wind speed in the upwind direction under HH polarization. This phenomenon arises because the simulation results of the conventional TSM based on the A spectrum always underestimate the scattering of Sentinel-1 owing to the neglect of wave breaking under HH polarization, and the positive correction introduced by the modified TSM in the upwind direction compensates for this underestimation to a certain extent. Furthermore, in the downwind direction under VV polarization, the NRCS calculated by the modified TSM based on the Tayfun ocean surface also performs better than the others because the TSM based on the A spectrum slightly overestimates the backscatter of Sentinel-1 data, and the negative correction introduced by the modified TSM compensates for some of the overestimation.

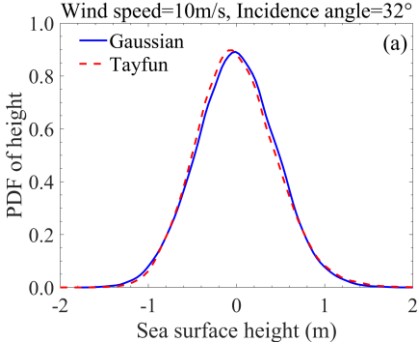
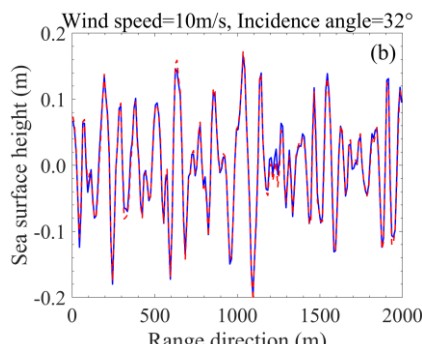
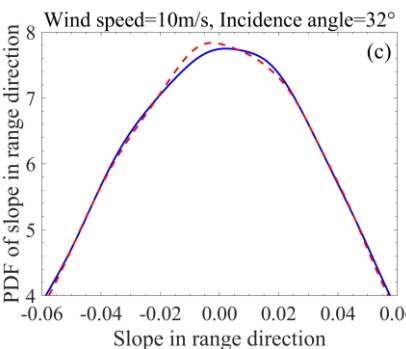

**Figure 7.** Comparison of a Gaussian sea surface and Tayfun sea surface at 10 m/s wind speed and 32° incidence angle in the upwind direction. (**a**) Pdfs for surface elevations; (**b**) profiles of surface elevations; (**c**) pdfs for slopes along range direction. Blue solid curve, Gaussian sea surface; red dashed curve, non-Gaussian sea surface constructed by Tayfun model.

The average NRCSs simulated based on the conventional TSM + Gaussian sea surface, conventional TSM + Tayfun sea surface, and modified TSM + Tayfun sea surface and their comparison with the reference data as functions of the wind direction at the wind speed of 10 m/s and incidence angle of 32° under VV and HH polarizations are shown in Figure 10. This figure shows that the simulated NRCS based on the conventional TSM + Gaussian sea surface is almost the same as that based on the conventional TSM + Tayfun sea surface over the wind direction range of 0°–360°. The modified TSM can simulate the upwind/downwind asymmetry of backscatter, which cannot be described by the conventional TSM. The modified TSM achieves this by introducing a positive correction in the upwind direction (0°) and a negative correction in the downwind direction (180°) using the complementary term proportional to the sea surface bispectrum. Moreover, the

minimum of the modified-TSM-simulated NRCS deviates from the crosswind direction (90°/270°) and moves to the downwind direction. All these changes diminish the gap between the predicted NRCS and the measured NRCS.

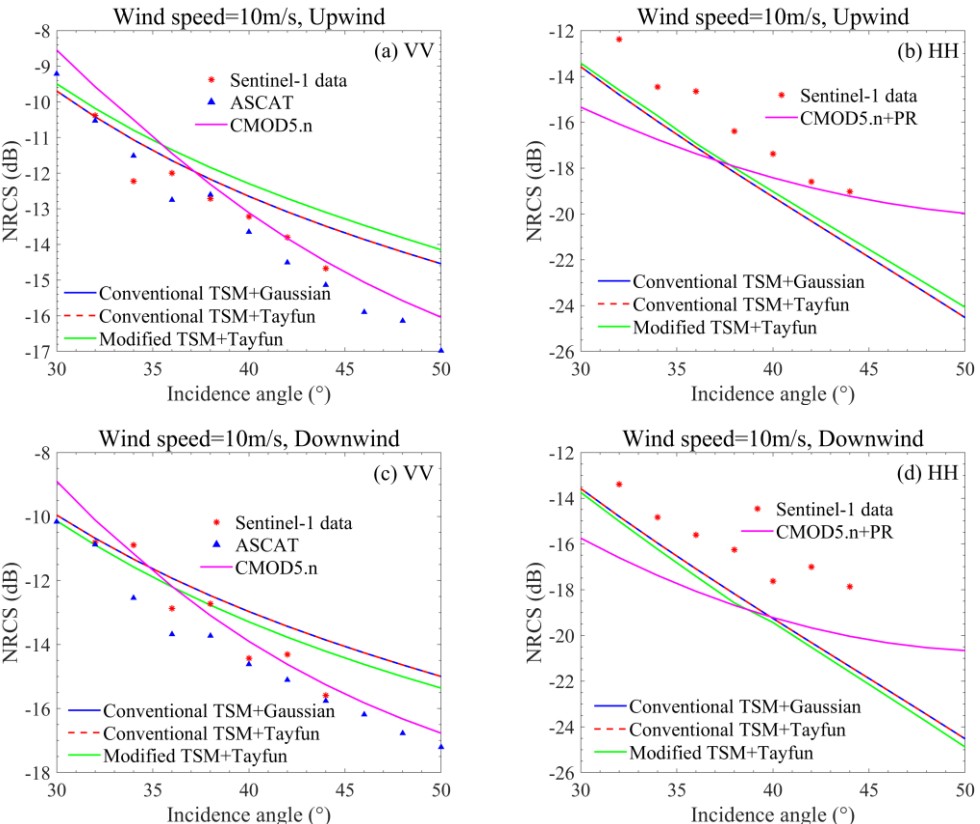

**Figure 8.** The simulated average NRCSs based on the conventional TSM + Gaussian sea surface, conventional TSM + Tayfun sea surface, and modified TSM + Tayfun sea surface and their comparison with the reference data at a 10 m/s wind speed as functions of the incidence angle (**a**,**b**) in the upwind direction under VV and HH polarizations and (**c**,**d**) in the downwind direction under VV and HH polarizations. Blue solid curve, the conventional TSM + Gaussian sea surface; red dashed curve, the conventional TSM + Tayfun sea surface; green solid curve, the modified TSM + Tayfun sea surface; red star, the Sentinel-1 data; blue triangle, the ASCAT; magenta solid curve, the GMF.

The upwind–downwind asymmetries predicted based on the conventional TSM + Gaussian sea surface, the conventional TSM + Tayfun sea surface, and the modified TSM + Tayfun sea surface and their comparison with the reference data as functions of the incidence angle at the wind speed of 10 m/s under VV and HH polarizations are shown in Figure 11a,b, and those as a function of wind speed at the incidence angle of 32° under VV and HH polarizations are shown in Figure 11c,d. The upwind–downwind asymmetry cannot be predicted by the conventional TSM based on either the Gaussian or the Tayfun sea surface, indicating that the construction of the Tayfun sea surface cannot explain the upwind–downwind asymmetry of sea surface scattering. The differences between the upwind and downwind NRCSs simulated by the modified TSM are in good agreement with the measurement data and the NRCSs predicted by the empirical model at the incidence angle of 30°–50° and a wind speed of 3 m/s–16 m/s under VV and HH polarizations, which is consistent with [36].

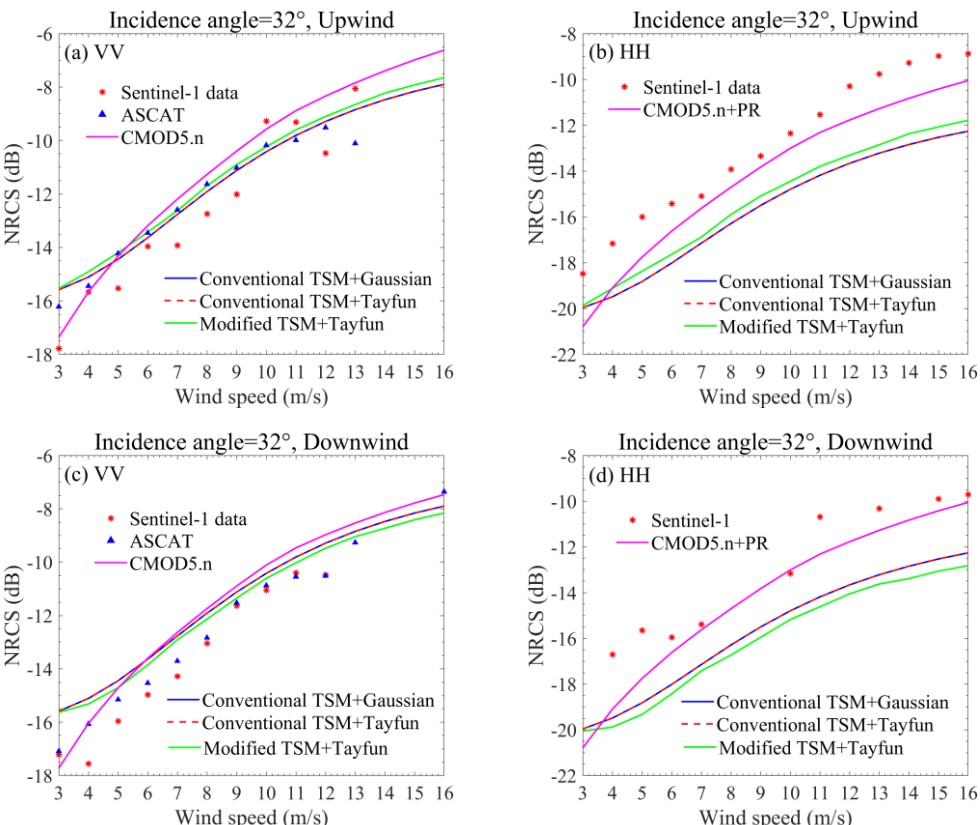

**Figure 9.** The simulated average NRCSs based on conventional TSM + Gaussian sea surface, conventional TSM + Tayfun sea surface, and modified TSM + Tayfun sea surface and their comparison with the reference data at a 32° incidence angle as functions of the wind speed (**a**,**b**) in the upwind direction under VV and HH polarizations and (**c**,**d**) in the downwind direction under VV and HH polarizations. Curves and symbols are the same as those in Figure 8.

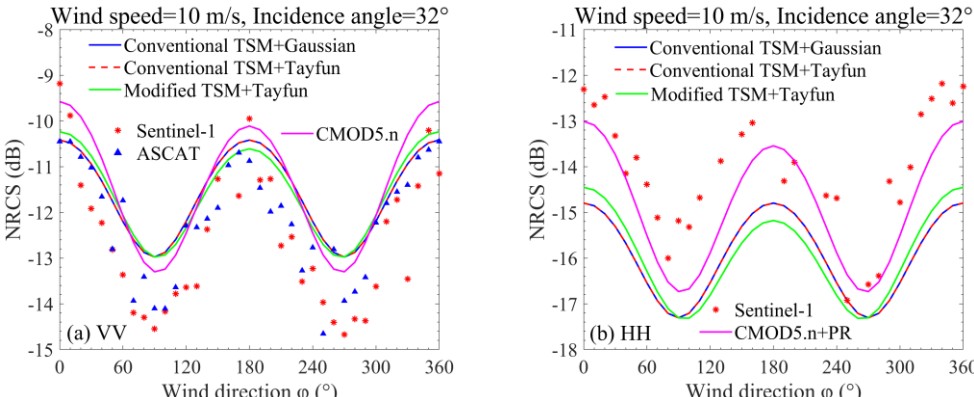

**Figure 10.** The average NRCSs based on conventional TSM + Gaussian sea surface, conventional TSM + Tayfun sea surface, and modified TSM + Tayfun sea surface and their comparison with reference data as functions of the wind direction at a 10 m/s wind speed and 32° incidence angle under VV and HH polarizations. Curves and symbols are the same as those in Figure 8. (**a**) The average NRCSs as functions of the wind direction at a 10 m/s wind speed and 32° incidence angle under VV polarization; (**b**) the average NRCSs as functions of the wind direction at a 10 m/s wind speed and 32° incidence angle under HH polarization.

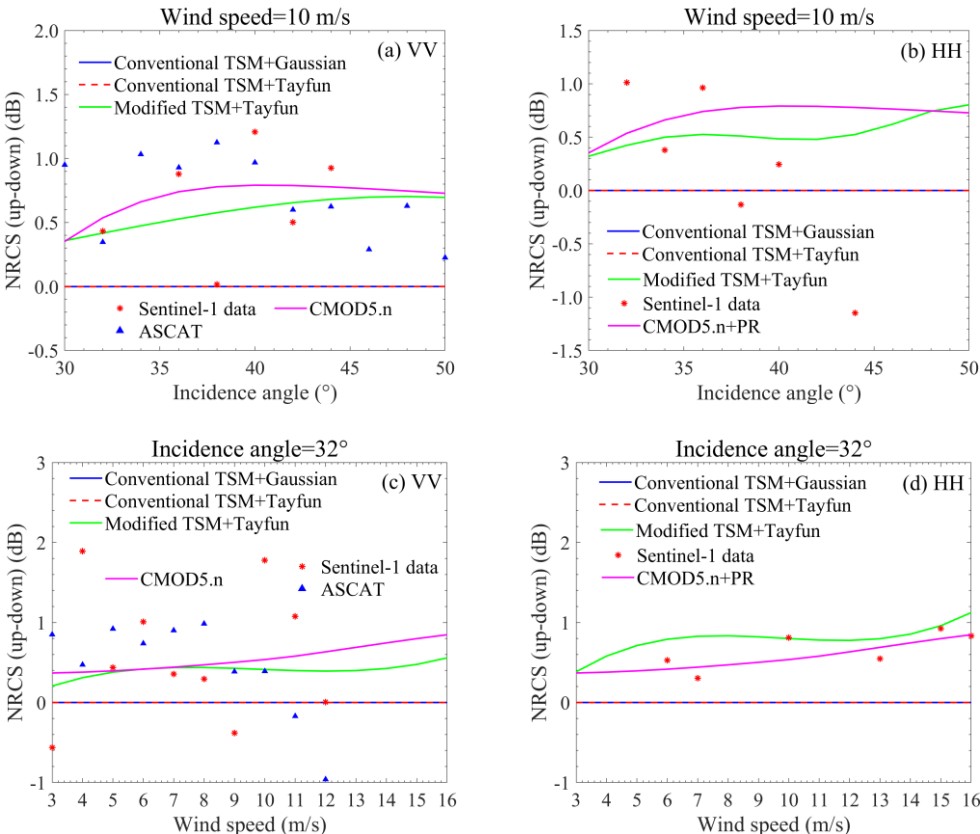

**Figure 11.** The upwind-downwind asymmetries predicted based on conventional TSM + Gaussian sea surface, conventional TSM + Tayfun sea surface, and modified TSM + Tayfun sea surface and their comparison with the reference data (**a**,**b**) as functions of the incidence angle at the wind speed of 10 m/s under VV and HH polarizations and (**c**,**d**) as functions of the wind speed at the incidence angle of 32° under VV and HH polarization. Curves and symbols are the same as those in Figure 8.

Examples of the pdfs for scattering predicted based on the conventional TSM + Gaussian sea surface, conventional TSM + Tayfun sea surface, and the modified TSM + Tayfun sea surface and their comparison with Sentinel-1 data and Gaussian fitting to Sentinel-1 data at the incidence angle of 32° and wind speed of 10 m/s in the upwind direction under VV and HH polarizations are shown in Figure 12a,b, and those in the downwind direction under VV and HH polarizations are shown in Figure 12c,d. The Sentinel-1 data in the IW mode from the wind wave sea surface were selected as the reference data, as shown in Figure 12. Notably, the IW Sentinel-1 data from the wind wave sea surface are minimal, especially in the downwind direction for VV polarization. Therefore, we appropriately widened the averaged bins of the Sentinel-1 data; that is, the Sentinel-1 data are averaged within ±0.7 m/s wind speed, ±7° wind direction, and ±0.7° incidence angle bins in the downwind direction under VV polarization. Figure 12 shows that the Sentinel-1 NRCS deviates from the Gaussian distribution, and the mean value shifts to a larger value in comparison with that of the Gaussian distribution. The introduction of a non-Gaussian large-scale sea surface based on the Tayfun model can make the scattering distribution deviate from the Gaussian distribution, but the deviation is very weak. The introduction of the bispectrum correction makes the scattering distribution deviate from the Gaussian distribution obviously, which is consistent with the Sentinel-1 data. As shown in Figure 12, the scattering predicted by the modified model shifts to a larger value compared with the conventional TSM in the upwind direction because of the positive correction introduced by the bispectrum. The results are opposite in the downwind direction. Figure 12 also shows that the extent of the Sentinel-1 NRCS distribution is always larger than that of the

simulated NRCS, especially under VV polarization. This might be caused by the noise contained in the Sentinel-1 data.

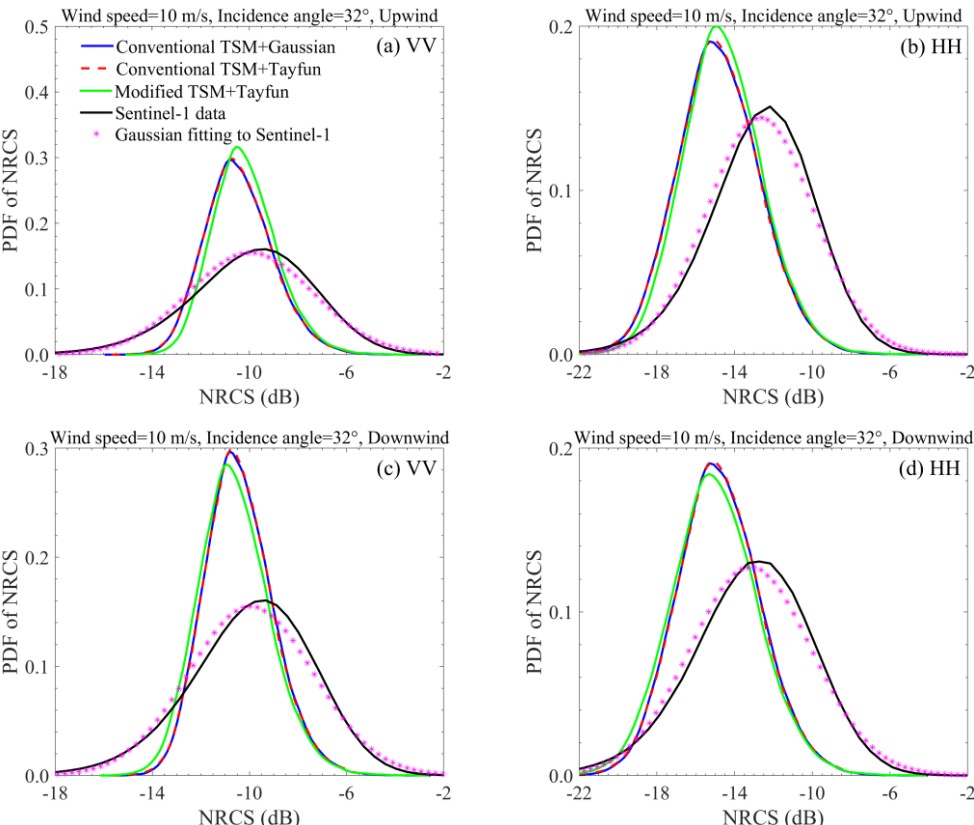

**Figure 12.** Examples of the pdfs for scattering predicted based on the conventional TSM + Gaussian sea surface, the conventional TSM + Tayfun sea surface, and the modified TSM + Tayfun sea surface and their comparison with Sentinel-1 data and Gaussian fitting to Sentinel-1 data at a 32° incidence angle and 10 m/s wind speed (**a,b**) in the upwind direction under VV and HH polarizations and (**c,d**) in the downwind direction under VV and HH polarizations. Blue solid curve, conventional TSM + Gaussian sea surface; red dashed curve, conventional TSM + Tayfun sea surface; green solid curve, modified TSM + Tayfun sea surface; black solid curve, Sentinel-1 data; magenta star, Gaussian distribution fitting to Sentinel-1 data.

Examples of the scattering profiles simulated based on the conventional TSM + Gaussian sea surface, the conventional TSM + Tayfun sea surface, and the modified TSM + Tayfun sea surface and their comparison with Sentinel-1 data at an incidence angle of 32° and wind speed of 10 m/s in the upwind direction under VV and HH polarizations are shown in Figure 13a,b, and those in the downwind direction under VV and HH polarizations are shown in Figure 13c,d. Figure 13 shows that the range of the fluctuation of the simulated NRCS is consistent with that of the reference data for a given wind direction and polarization at a certain wind speed and certain incidence angle. Neither the introduction of the non-Gaussianity of the Tayfun sea surface nor the introduction of the bispectrum correction substantially affects the range of fluctuation of the simulated scattering.

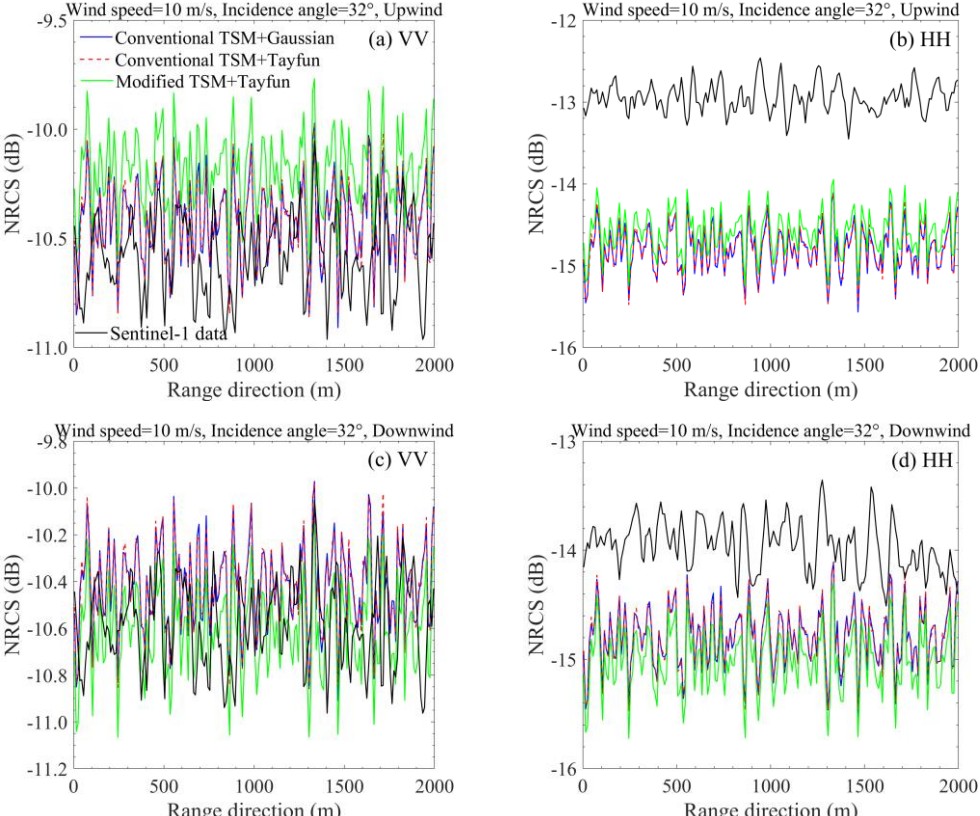

**Figure 13.** Examples of the scattering profiles predicted based on the conventional TSM + Gaussian sea surface, the conventional TSM + Tayfun sea surface, and the modified TSM + Tayfun sea surface and their comparison with Sentinel-1 data at 32° incidence angle and 10 m/s wind speed (**a**,**b**) in the upwind direction under VV and HH polarizations and (**c**,**d**) in the downwind direction under VV and HH polarizations. Blue solid curve, conventional TSM + Gaussian sea surface; red dashed curve, conventional TSM + Tayfun sea surface; green solid curve, modified TSM + Tayfun sea surface; black solid curve, Sentinel-1 data.

*3.3. Effect of Swell on NRCS Simulation*

The swell can change the local height fluctuation and thus affect the patterns of the facet scattering [44]. However, research on the influence of swell on the distribution of facet scattering is lacking. In this section, we evaluate the effect of swell on the sea surface height distribution and TSM-simulated scattering distribution at the incidence angles of 30°–50° and wind speeds of 3–16 m/s under VV and HH polarizations.

Figure 14 shows a comparison example of the distributions of the Tayfun sea surface without and with swell at the wind speed of 10 m/s and incidence angle of 32° in the upwind direction. Here, the significant wave height of swell = 4 m, swell wavelengths = 200 m, and the swell direction is upwind. Figure 14a shows that the introduction of swell has minimal impact on the mean sea surface height but widens the range of the elevation distribution of the sea surface; that is, the sea surface with swell has a lower probability near small heights and higher probability near large heights than that of the sea surface without swell. Figure 14b shows that the range of fluctuation of sea surface height with swell is substantially larger than that of the wind wave surface without swell. Figure 14c indicates that the probability of the sea surface slope with swell is higher at large slopes and lower at small slopes in comparison with that of the sea surface without swell. Additionally, we repeated the experiments at a low wind speed (i.e., 5 m/s) and found that the derived results are similar to those achieved at a wind speed of 10 m/s, while the difference between the sea surface slope with and without swell is more obvious at a low wind speed.

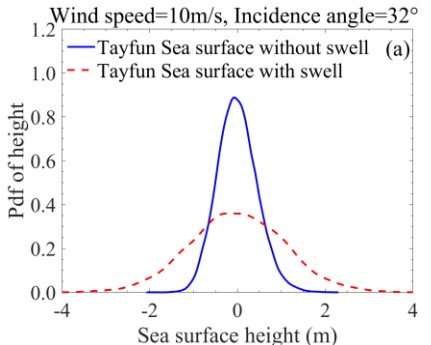
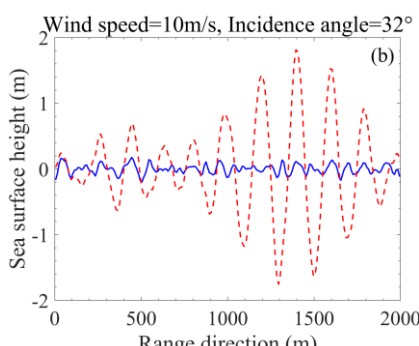
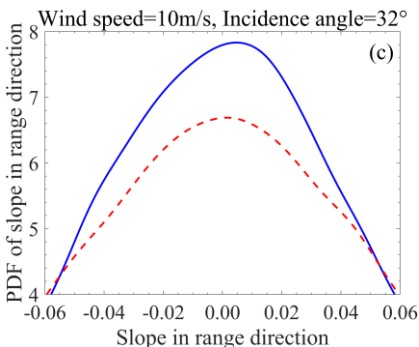

**Figure 14.** Comparison of Tayfun sea surface without and with swell at 10 m/s wind speed and 32° incidence angle in the upwind direction. The significant wave height of swell = 4 m, swell wavelengths = 20 m, and the swell direction is upwind. (**a**) Pdfs for surface elevations; (**b**) profiles of surface elevations along range direction; (**c**) pdfs for slopes along range direction. Blue solid curve, sea surface without swell; red dashed curve, sea surface with swell.

Figure 15 shows two examples of the scattering pdfs predicted by the modified TSM based on the Tayfun sea surface without and with swell and their comparison with the Sentinel-1 data. The detailed parameters of the examples are as follows: in Figure 15a, VV polarization, incidence angle of 35°, wind speed of 13 m/s, wind direction of 240° relative to the range direction, significant wave height of swell = 4 m, swell wavelengths = 200 m, and the direction of the dominant wave is 180° relative to the range direction; in Figure 15b, HH polarization, incidence angle of 44°, wind speed of 7.4 m/s, wind direction of 91° relative to the range direction, significant wave height of swell = 4 m, swell wavelengths = 196 m, and the direction of the dominant wave is 191° relative to the range direction. We can see from Figure 15 that the introduction of swell widens the range of the scattering distribution, and the peak value of the NRCS pdf from the sea surface with swell is lower than that without swell. All these changes diminished the gap between the simulated NRCS and the measured NRCS. Figure 16 shows the profile of NRCS in the range direction based on the same parameters in Figure 15. It can be seen from Figure 16 that the introduction of swell widens the range of fluctuation of the NRCS profile and that it is in better agreement with the Sentinel-1 data than that of the wind wave sea surface without swell.

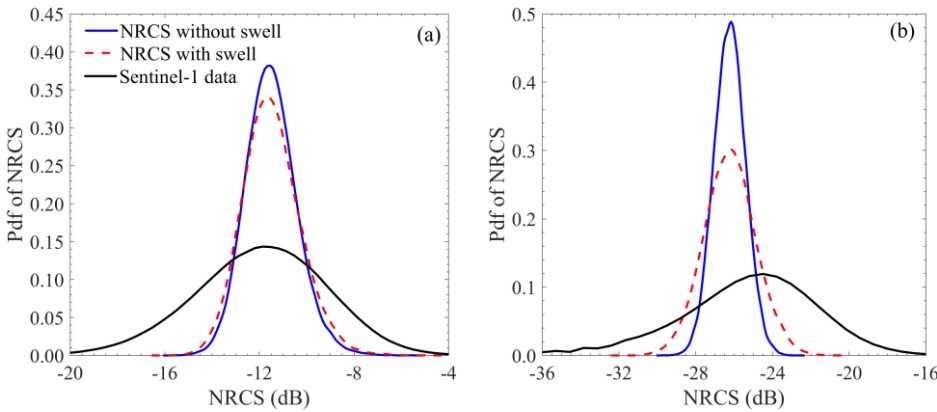

**Figure 15.** Two examples of scattering pdfs predicted by the modified TSM from Tayfun sea surface without and with swell and their comparison with the Sentinel-1 data. (**a**) Incidence angle is 35°, wind speed is 13 m/s, wind direction is 240° relative to range direction, VV polarization, significant wave height of swell is 4 m, swell wavelength is 200 m, and direction of the dominant wave is 180° relative to range direction; (**b**) incidence angle is 44°, wind speed is 7.4 m/s, wind direction is 91° relative to range direction, HH polarization, significant wave height of swell is 4 m, swell wavelength is 196 m, and direction of the dominant wave is 191° relative to range direction.

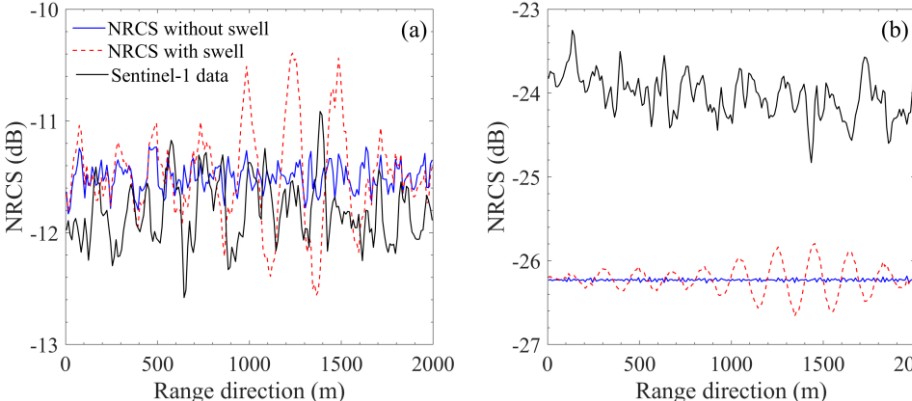

**Figure 16.** The profile of NRCS in the range direction based on the same parameters in Figure 15. (**a**) Incidence angle is 35°, wind speed is 13 m/s, wind direction is 240° relative to range direction, VV polarization, significant wave height of swell is 4 m, swell wavelength is 200 m, and direction of the dominant wave is 180° relative to range direction; (**b**) incidence angle is 44°, wind speed is 7.4 m/s, wind direction is 91° relative to range direction, HH polarization, significant wave height of swell is 4 m, swell wavelength is 196 m, and direction of the dominant wave is 191° relative to range direction.

To further analyze the impact of swell, we used three examples of simulated scattering based on the modified TSM + Tayfun sea surface without and with swell and their comparison with the Sentinel-1 data at low, medium, and high wind speeds in two respects: the scattering image and the scattering image spectrum [46]. The detailed parameters of three examples are shown in Table 1. Here, the size and resolution of the sea surface were set as 5 km × 5 km and 10 m × 10 m, respectively.

**Table 1.** The parameters of examples.

| ID | Incidence Angle (degree) | Wind Speed (m/s) | Wind Direction (degree) | Significant Wave Height (m) | Wavelength of the Dominant Wave (m) | Direction of the Dominant Wave (degree) |
|----|----|----|----|----|----|----|
| 1 | 32.2 | 4.7 | 260.0 | 2.0 | 171.5 | 149.0 |
| 2 | 41.5 | 7.1 | 137.7 | 2.6 | 185.7 | 158.2 |
| 3 | 35.0 | 13.0 | 240.0 | 4.1 | 200.0 | 180.0 |

Figure 17 shows the scattering images corresponding to ID 1 of (a) the Sentinel-1 data, (b) simulated NRCS based on modified TSM + Tayfun sea surface without swell, (c) the area enclosed by the red solid line in (b), and (d) simulated NRCS based on modified TSM + Tayfun sea surface with swell. We can see that, at a low wind speed (4.7 m/s), the simulated NRCS based on the wind wave ocean surface without swell shows only the textural features of the wind waves, and the patterns of the swell are not displayed. A comparison of the scattering patterns with swell (i.e., Figure 17d) with the Sentinel-1 data (i.e., Figure 17a) revealed that the textural features in the actual image were simulated successfully when swell is considered.

To quantitatively compare the consistency of the simulated textural features with the Sentinel-1 data, Figure 18 presents the image spectra of the scattering image corresponding to ID 1 of (a) Sentinel-1 data, (b) simulation based on the modified TSM + Tayfun sea surface without swell, and (c) simulation based on the modified TSM + Tayfun sea surface with swell. Furthermore, the wavelengths of the dominant wave and the directions of the dominant wave obtained from the image spectra in Figure 18 are listed in Table 2. The dominant wave wavelength and dominant wave direction of the simulation without swell are different from those of the Sentinel-1 data, and the spectral density of the simulation without swell is much smaller than that of the Sentinel-1 data. This is because only wind

wave information such as wind direction and wavelength is contained in the scattering image. In comparison, the dominant wave wavelength and dominant wave direction of the simulation with swell are in good agreement with the Sentinel-1 data, with errors of $\pm 0.1$ m and $\pm 0.5°$, respectively. Moreover, the introduction of swell diminishes the gap between the spectral density of the simulation and the measurement.

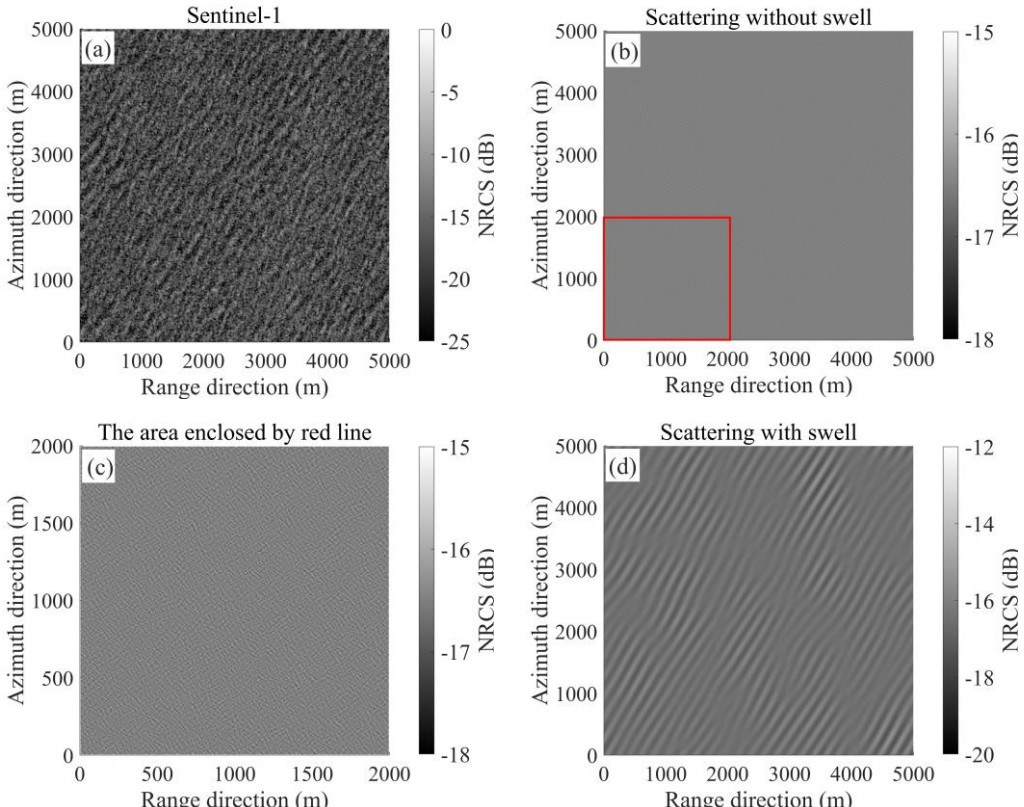

**Figure 17.** The scattering images corresponding to ID 1 of (**a**) the Sentinel-1 data, (**b**) simulation based on modified TSM + Tayfun sea surface without swell, (**c**) the area enclosed by red solid line in (**b**), and (**d**) simulation based on modified TSM + Tayfun sea surface with swell.

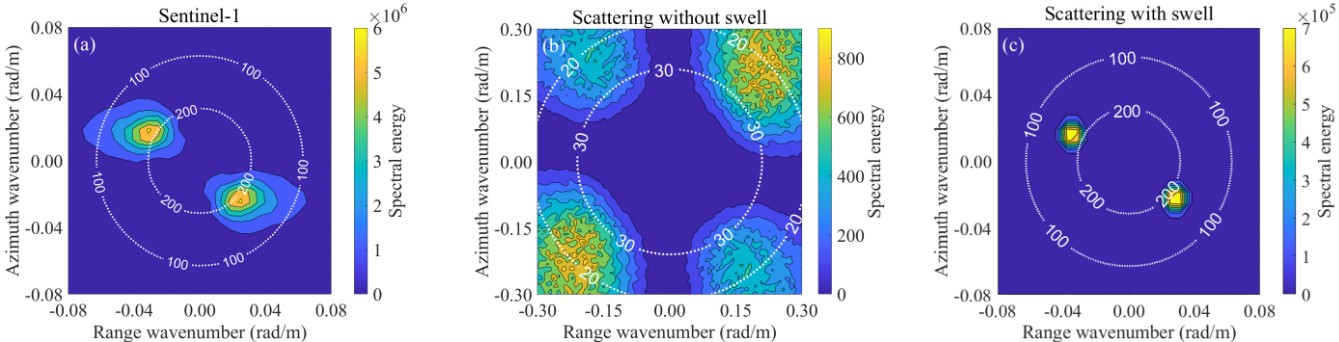

**Figure 18.** The scattering image spectra corresponding to ID 1 of (**a**) the Sentinel-1 data, (**b**) simulation based on modified TSM + Tayfun sea surface without swell, and (**c**) simulation based on modified TSM + Tayfun sea surface with swell.

**Table 2.** The wavelength and direction of dominant wave obtained from the scattering image spectra in Figure 18.

| Ocean Surface | Wavelength of the Dominant Wave (m) | Direction of the Dominant Wave (degree) |
|---|---|---|
| Sentinel-1 data | 171.5 | 149.0 |
| Sea surface without swell | 19.4 | 227.4 |
| Sea surface with swell | 171.5 | 149.1 |

Figure 19 shows the scattering images corresponding to ID 2. This figure shows that the patterns of the scattering image of the wind wave sea surface at moderate wind speed (e.g., 7.1 m/s) are clearer than those at a low wind speed (e.g., 4.7 m/s) because the wavelength of wind waves increases with the wind speed. The direction of the stripes in the simulated scattering image without swell is similar to that of the Sentinel-1 data because the direction of the wind wave propagation is similar to that of swell in this example. The patterns of the Sentinel-1 data are well simulated by the simulation with swell.

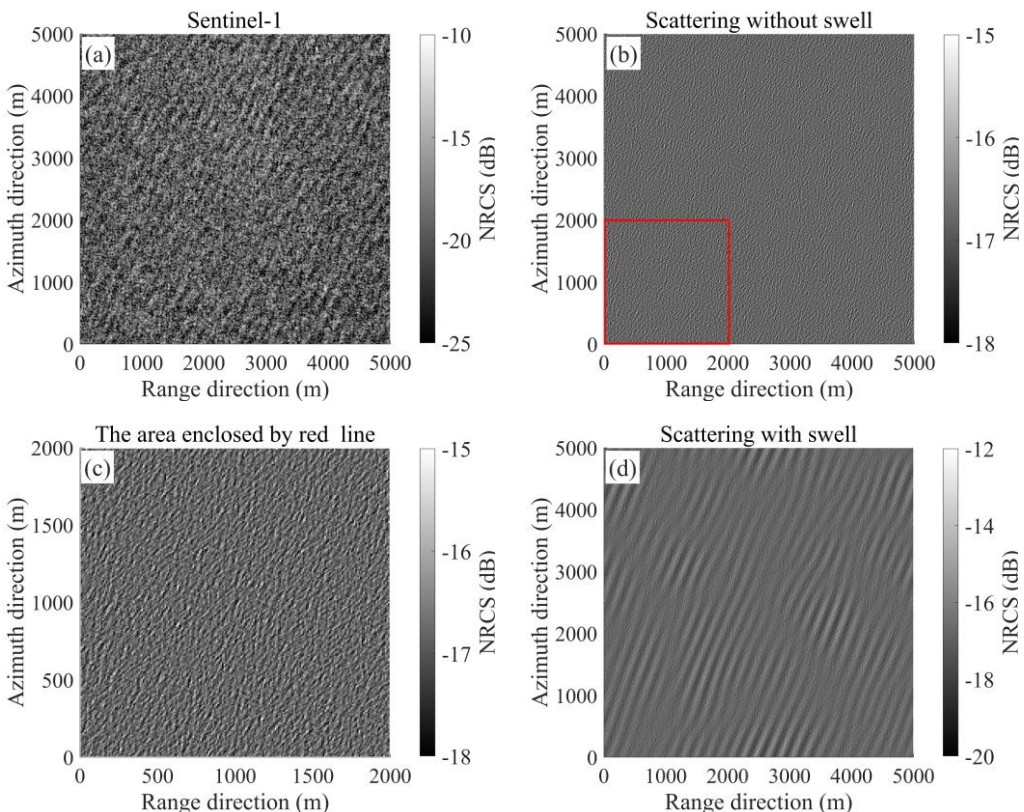

**Figure 19.** Similar to Figure 17 but corresponding to ID 2.

Figure 20 presents the image spectra of the scattering image corresponding to ID 2. The wavelengths and directions of the dominant waves obtained from the scattering image spectra in Figure 20 are listed in Table 3. The level of the simulated spectrum based on the wind wave sea surface at a moderate wind speed was determined to be larger than that at a low wind speed, but a large difference remains between the simulated spectral density and the measurement, and the introduction of swell diminishes the gap. As expected, the wavelength and direction of the dominant waves simulated using the sea surface with swell are almost the same as those of the Sentinel-1 data.

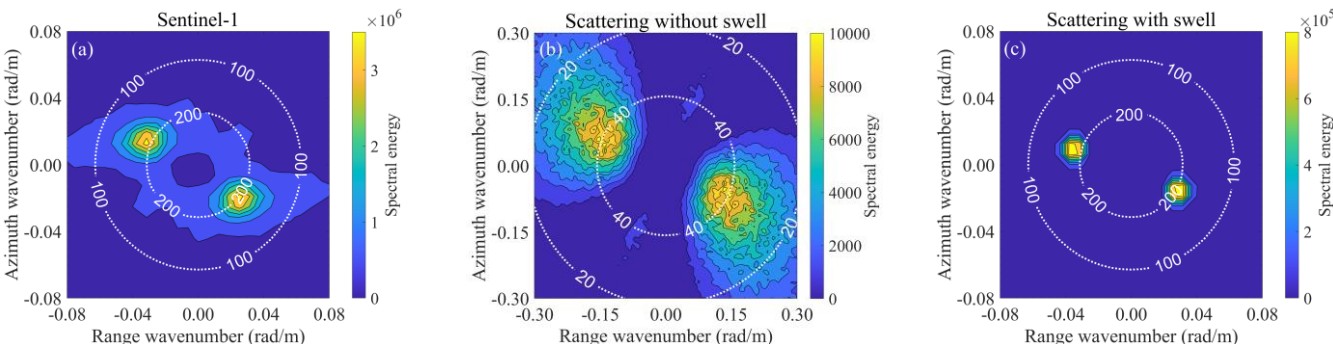

**Figure 20.** Similar to Figure 18 but corresponding to ID 2.

**Table 3.** The wavelength and direction of dominant wave obtained from the image spectra in Figure 20.

| Ocean Surface | Wavelength of the Dominant Wave (m) | Direction of the Dominant Wave (degree) |
|---|---|---|
| Sentinel-1 data | 185.7 | 158.2 |
| Sea surface without swell | 44.5 | 150.1 |
| Sea surface with swell | 185.7 | 158.2 |

The scattering images and scattering image spectra corresponding to ID 3 are shown in Figures 21 and 22, respectively. The wavelengths and directions of the dominant waves obtained from the image spectra in Figure 22 are listed in Table 4. A comparison of the simulation and Sentinel-1 data revealed that the scattering image patterns and image spectrum simulated based on the sea surface with swell are consistent with those of the actual Sentinel-1 data and that the errors of the simulation can be ignored.

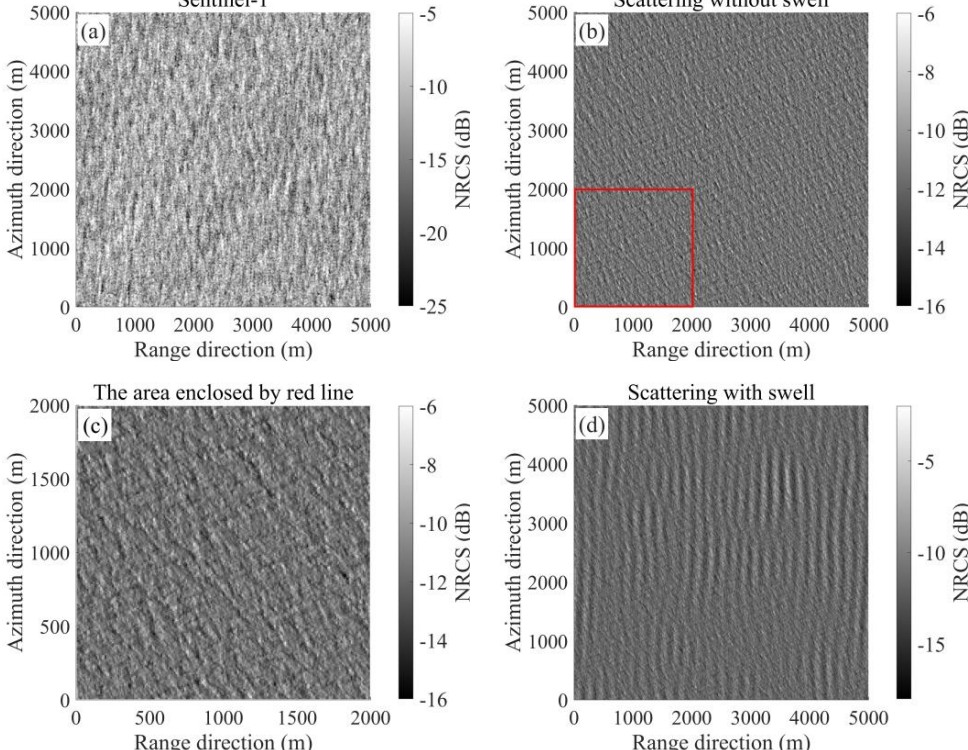

**Figure 21.** Similar to Figure 17 but corresponding to ID 3.

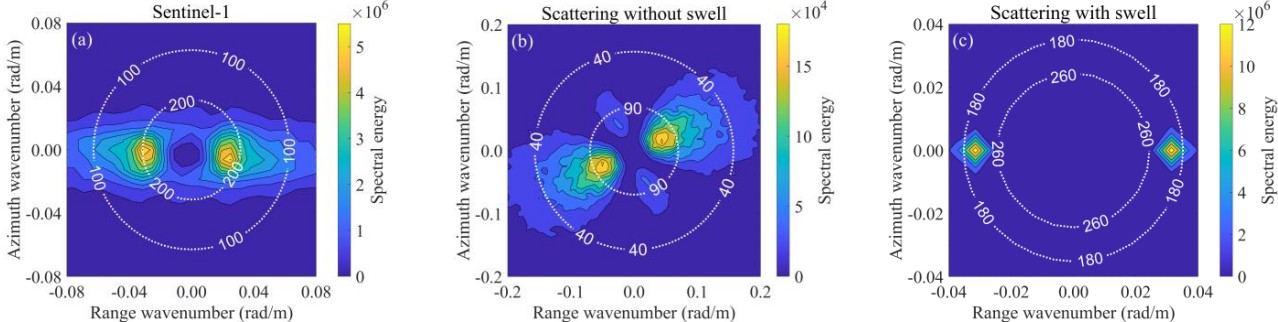

**Figure 22.** Similar to Figure 18 but corresponding to ID 3.

**Table 4.** The wavelength and direction of dominant wave obtained from the image spectra in Figure 22.

| Ocean Surface | Wavelength of the Dominant Wave (m) | Direction of the Dominant Wave (degree) |
| --- | --- | --- |
| Sentinel-1 data | 200.0 | 180.0 |
| Tayfun sea surface without swell | 131.3 | 203.2 |
| Tayfun sea surface with swell | 200.0 | 180.0 |

As highlighted above, the scattering image patterns and scattering image spectra of Sentinel-1 data can be successfully simulated by the modified TSM based on the Tayfun ocean surface with swell at various sea states. In [46], it is also proved that the textural features in simulated SAR images based on swell are in good agreement with those of the RADARSAT-2 SAR image.

## 4. Discussion

*Effect of Ocean Parameters on the NRCS Simulation*

The parameters of the sea surface have a substantial impact on the efficiency of scattering calculation. Therefore, we analyze the influence of ocean parameters on both the scattering prediction and the computational efficiency based on the conventional TSM + Gaussian sea surface using the A spectrum and cosine spreading function in this section.

The effects of the sea surface size on the prediction of backscattering are explored in detail in this section. We found that the size of the ocean surface area has little effect on the scattering distribution and the average NRCS at different wind speeds under various incidence angles. However, the computation time increases dramatically as the surface size increases. An example based on a Gaussian sea surface with areas of 2 km × 2 km and 10 km × 10 km, facet size of 5 m × 5 m, wind speed of 10 m/s, and incidence angle of $40°$ is taken to demonstrate the influence of the sea surface size on the NRCS simulation. The pdfs for height, slope, and scattering in this example are shown in Figure 23, and the scattering images associated with this example are shown in Figure 24. The range and azimuth slope were calculated in the upwind and crosswind directions, respectively. Figure 23 shows almost no difference in the pdfs for the sea surface elevation and slope under various ocean sizes. Moreover, the surface size has no impact on the scattering distribution. From Figure 24, the patterns of the scattering images with different sea surface sizes can be determined to be almost identical at the same scale. A comparison of the average NRCS based on different sea surface sizes revealed that the mean scattering from the sea surface with an area of 2 km × 2 km is almost the same as that from the sea surface with an area of 10 km × 10 km for a given wind speed, incidence angle, and facet size. For example, the NRCS from the sea surface with an area of 2 km × 2 km and that from the sea surface with an area of 10 km × 10 km under VV polarization is $-15.24$ dB when

the wind direction is upwind, the wind speed is 10 m/s, and the incidence angle is 40°. However, the calculation time increases considerably when the sea surface size is increased. The experiments outlined above were coded in MATLAB and run on a PC with an Intel(R) Xeon(R) Gold 6130 CPU. The code required approximately 30 min to obtain the NRCS for a single wind speed and incidence angle for a sea surface with an area of 2 km × 2 km; in comparison, a period of 4320 min was required to calculate the backscatter for a sea surface with an area of 10 km × 10 km under the same conditions.

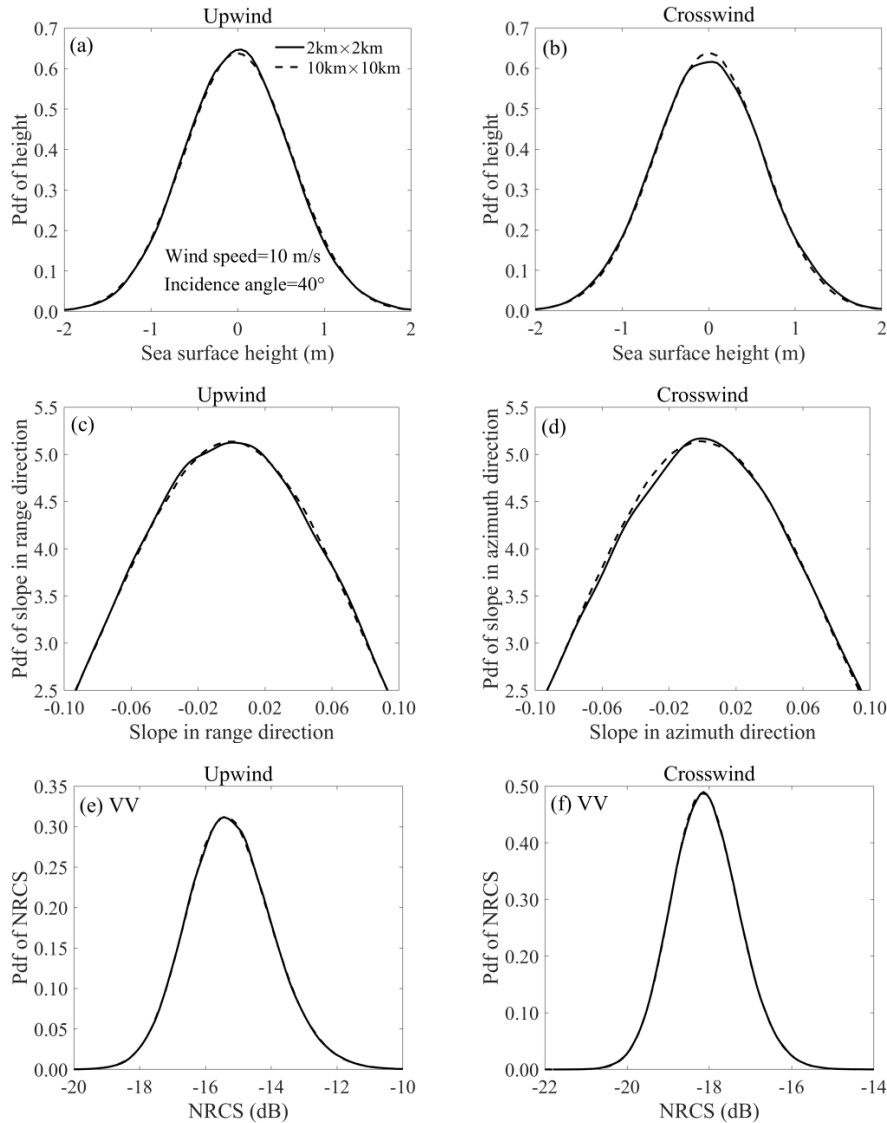

**Figure 23.** Comparison of the distributions of sea surface with different ocean surface sizes at the incidence angle of 40° and wind speed of 10 m/s in the upwind (**left**) and crosswind (**right**) directions. The facet resolution is 5 m × 5 m. (**a**,**b**) Pdfs for elevation; (**c**,**d**) pdfs for slope; (**e**,**f**) pdfs for backscatter under VV polarization. Solid curves, the size of the sea surface is 2 km × 2 km; dashed curves, the size of the sea surface is 10 km × 10 km.

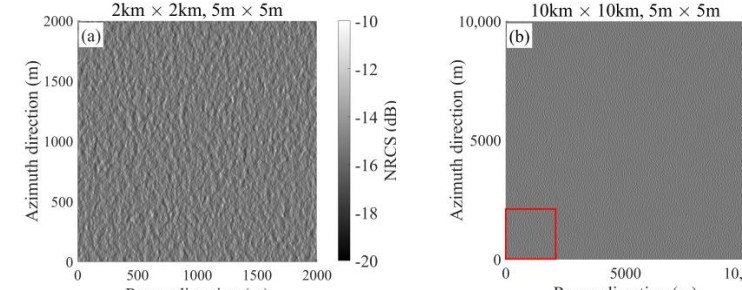
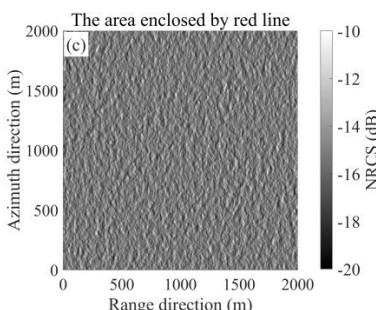

**Figure 24.** Comparison of scattering images based on different ocean surface sizes at the incidence angle of 40° and wind speed of 10 m/s in the upwind direction under VV polarization. The facet resolution is 5 m × 5 m. (**a**) Sea surface in the area of 2 km × 2 km; (**b**) sea surface in the area of 10 km × 10 km; (**c**) the area enclosed by red solid line in (**b**).

The impact of the facet size on the NRCS was analyzed for a specific sea surface area. We found that the average NRCS, scattering profile, and scattering pattern are not considerably affected by a change in facet size from 1 to 10 m. However, the pdfs for backscatter are different depending on the facet size. More time is required to calculate the scattering from the sea surface with a smaller facet size. An example based on a Gaussian sea surface with an area of 2 km × 2 km and facet sizes of 1 m × 1 m, 5 m × 5 m, and 10 m × 10 m under 10 m/s wind speed at an incidence angle of 40° was used to demonstrate the influence of sea surface size on the NRCS simulation. The pdfs for the height, slope, and scattering of this example are shown in Figure 25. Figure 26 presents the profiles of wave elevation and the NRCS of this example, and Figure 27 shows the scattering images. From Figure 25, we can see that, for a larger facet size, the height pdf, slope pdf, and NRCS pdf are larger around the mean value and lower further from the mean value. Figure 26 shows that the ranges of fluctuation of the scattering profiles based on different sea surface resolutions are similar. Figure 27 shows that the impact of facet size on the scattering pattern is small and can be ignored to a certain extent. The analysis of the effect of sea surface resolution on the mean scattering reveals only a slight difference, which can be ignored between the average scatterings based on different facet sizes. For example, the NRCSs of the sea surface with facet sizes of 1, 5, and 10 m are −15.26, −15.24, and −15.23 dB, respectively. This result is consistent with the conclusion obtained in [10] that the average does not considerably depend on the facet size. The times required to obtain the scattering with facet sizes of 1, 5, and 10 m are 1, 30, and 4320 min, respectively, when the sea surface size is 2 km × 2 km. In subsequent experiments, we chose a facet size of 10 m to construct the sea surface, which is the same as the resolution of Sentinel-1 IW high-resolution data.

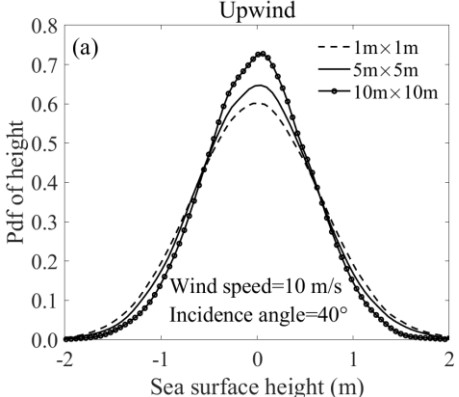
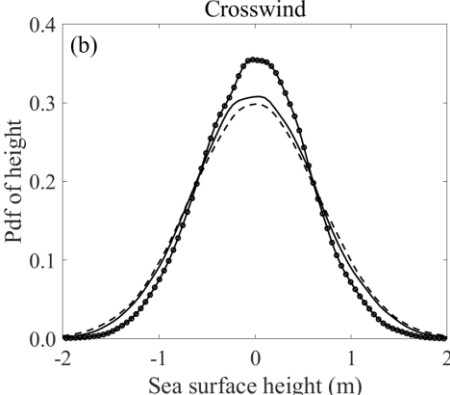

**Figure 25.** *Cont*.

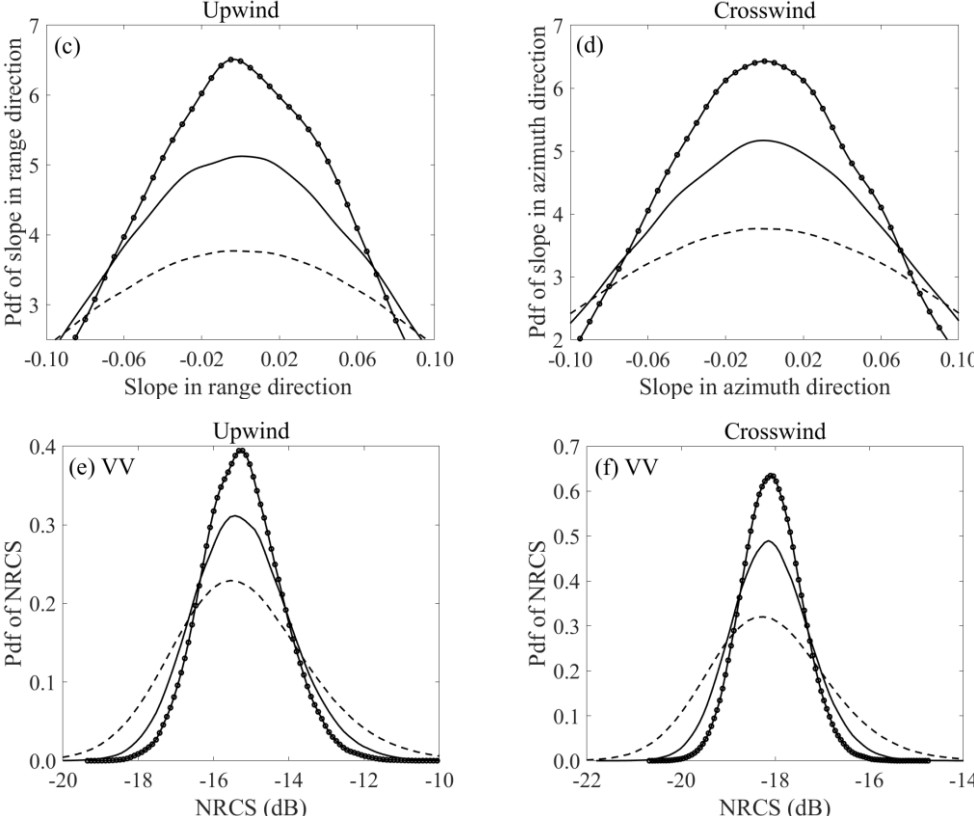

**Figure 25.** Comparison of the distributions of sea surface with different facet sizes in the upwind (**left**) and crosswind (**right**) directions at the incidence angle of 40° and wind speed of 10 m/s. The surface size is 2 km × 2 km. (**a,b**) Pdfs for elevation; (**c,d**) pdfs for slope; (**e,f**) pdfs for backscatter under VV polarization. Dashed curves, facet size is 1 m × 1 m; solid curves, facet size is 5 m × 5 m; solid curves with dot, facet size is 10 m × 10 m.

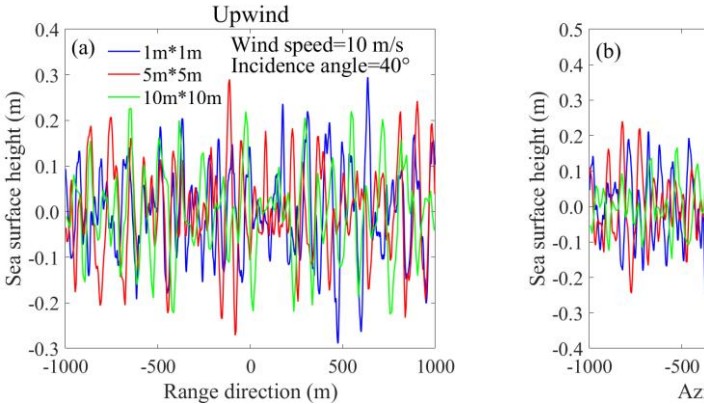

**Figure 26.** *Cont.*

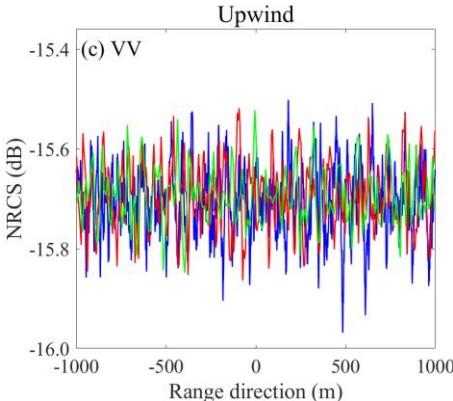
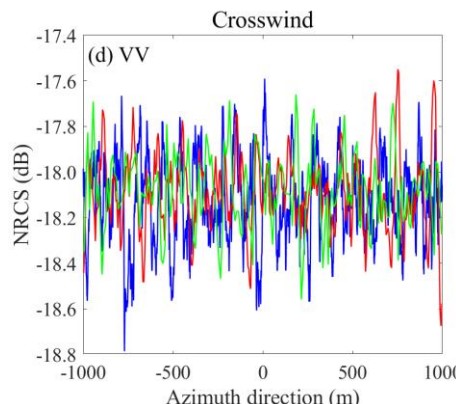

**Figure 26.** Comparisons of profiles of the sea surface height and profiles of scattering intensity under VV polarization based on different facet sizes in the upwind (**left**) and crosswind (**right**) directions at the incidence angle of 40° and wind speed of 10 m/s. The surface size is 2 km × 2 km. Blue solid curves, 1 m × 1 m; red solid curves, 5 m × 5 m; green solid curves, 10 m × 10 m. (**a**) Profiles of sea surface heights in the upwind direction; (**b**) profiles of sea surface heights in the crosswind direction; (**c**) profiles of NRCSs in the upwind direction; (**d**) profiles of NRCSs in the crosswind direction.

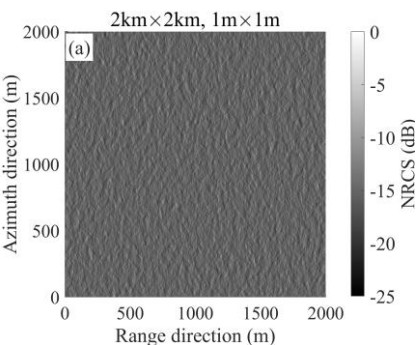
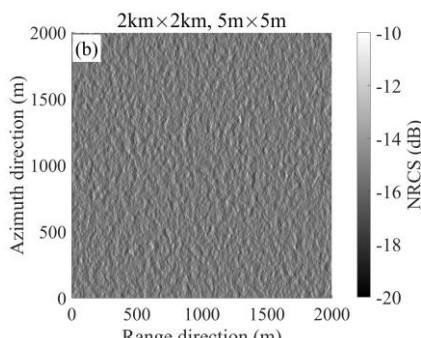
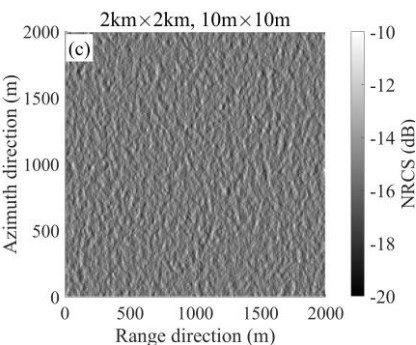

**Figure 27.** Comparisons of the scattering images based on different facet sizes in the upwind direction at the incidence angle of 40° and wind speed of 10 m/s under VV polarization. The surface size is 2 km × 2 km. (**a**) The facet size is 1 m × 1 m; (**b**) the facet size is 5 m × 5 m; (**c**) the facet size is 10 m × 10 m.

## 5. Conclusions

Most previous studies of facet scattering were carried out based on a Gaussian sea surface constructed using the E spectrum or JONSWAP spectrum [6–14]. The effects of different wind wave spectra and the non-Gaussianity of the sea surface on the average scattering and scattering distribution need to be further analyzed under different incidence angles, wind speeds, and wind directions. Meanwhile, most studies focused on the influence of swell on the mean NRCS, and there are few studies that analyze the impact of swell on facet scattering. Therefore, in this paper, we analyzed the effects of the wind wave spectrum, non-Gaussianity of the sea surface, and swell on the prediction of facet scattering based on the facet TSM by comparing the simulated results with the Sentinel-1 data, the ASCAT data, and the GMF at the wind speed range of 3–16 m/s, the wind direction range of 0°–360°, and the incidence angle range of 30°–50° under VV and HH polarizations.

First, we compared the performances of the TSM based on the four wind wave spectra of D, A, E, and H18 and the two composite spectra of AE and AEH18 in predicting mean scattering. The results show that the prediction based on the A spectrum is more consistent with the Sentinel-1 data and GMF under HH polarization. Under VV polarization, the A spectrum has an overall satisfactory performance under different incidence angles compared to the Sentinel-1 data, the ASCAT data, and the GMF, except for high incidence angles (e.g., > 35°), where the combined spectra performed slightly better. Therefore, the

A spectrum was found to have a better applicability generally. This result is different from the conclusion in [39] that a composite wind wave spectrum has a better applicability than single wave spectra. This might be because in [39], the influences of the spectra on the NRCS prediction were analyzed based on the advanced integral equation model. In addition, we compared the distribution of the simulated scattering using TSM based on different spectra with the Sentinel-1 data in the IW mode, and we found that the spread of the Sentinel-1 scattering distribution was wider than that of the simulation, which may be because the Sentinel-1 data contain noise. The A spectrum achieved a more consistent scattering distribution with the Sentinel-1 data at low incidence angles, while the composite spectrum performed better at high incidence angles under VV polarization.

Second, we analyzed the influence of the non-Gaussianity of the sea surface on the mean NRCS simulated by TSM based on the A spectrum. The results show that the upwind–downwind asymmetry of backscattering can be predicted well by the modified TSM which is constructed by incorporating bispectrum correction into the conventional TSM. This is consistent with the result reported in [45]. We also analyzed the influence of non-Gaussianity on the distribution of facet scattering and found that the introduction of a non-Gaussian large-scale sea surface constructed by the Tayfun model can make the scattering distribution deviate from the Gaussian distribution, but the deviation is very weak. In comparison, the distribution of the scattering with bispectrum correction deviates from the Gaussian distribution significantly, which is in good agreement with the Sentinel-1 data.

Third, we analyzed the influence of swell on the scattering distribution simulated by the modified TSM. The results show that the introduction of swell widened the spread of the scattering distribution, and the peak value of the NRCS pdf with swell was smaller than that without swell. Additionally, the range of fluctuation of the NRCS profile with swell was larger than that without swell. All these changes caused by the introduction of swell made the distribution of the TSM-simulated facet scattering more consistent with the Sentinel-1 data. At the same time, the scattering image patterns and scattering image spectrum of Sentinel-1 data were successfully simulated at various sea states when the swell was considered. Similar results can be found in [46], where the textural features of the simulated SAR images with the consideration of swell were in good agreement with those of the RADARSAT-2 SAR images.

**Author Contributions:** Conceptualization, C.F. and Q.Y.; methodology, Y.W.; software, Y.W.; validation, T.S., J.M. and J.Z.; writing—original draft preparation, Y.W.; writing—review and editing, C.F and Q.Y.; funding acquisition, J.Z. All authors have read and agreed to the published version of the manuscript.

**Funding:** This work was supported by the National Natural Science Foundation of China under Grant 61931025, by the Key Program of Joint Fund of the National Natural Science Foundation of China and Shandong Province under Grant U2006207, and by the Fundamental Research Funds for the Central Universities under Grant 21CX06015A.

**Institutional Review Board Statement:** Not applicable.

**Informed Consent Statement:** Not applicable.

**Data Availability Statement:** Not applicable.

**Acknowledgments:** The authors would like to thank the European Space Agency for providing the Sentinel-1 data and the EUMETSAT for providing electronic access to the ASCAT data. The authors would also like to thank NDBC for providing the buoy data.

**Conflicts of Interest:** The authors declare no conflict of interest.

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
