# Peer review of "Effects of Wind Wave Spectra, Non-Gaussianity, and Swell on the Prediction of Ocean Microwave Backscatter with Facet Two-Scale Model"

_remotesensing, doi:10.3390/rs15051469_

Round 1
Reviewer 1 Report
In this paper, various aspects/properties of the sea surface and wind are analyzed as important factors in the determination of the SAR scattering off the sea surface using a developed facet two-scale model, in comparison with three types of other data, including sat measurements data. The paper has a clear list of adequate references, to which the reader may turn if necessary, but the entire structure of the paper needs an update - some parts of the text are highlighted in blue with minor errors in editing (a remnant of older editing, or an incorrect PDF document creation?).
Lines 61-62 - JONSWAP & E spectrums - a short (for few wors) explanation would be beneficiary; E spectrum is explained in 143-144, but this should appear earlier, with the first mention of the 'E spectrum'.
The addressed subject is very important, which is clearly shown in the discussion (exemplary data). Very thorough descriptions of the theorized and simulated mechanisms, which provides a high amount of discussed detail and ensured repeatability of the work in the future/other projects. The effect of swell has been well defined and its effect has been clearly shown on the examples. The Discussion is pertinent.
Good description of the spectra (Fig. 1).
Fig. 2 - the details addressed by the text are not clearly shown (low quality, lower than e.g. Fig. 18).
Figures 1, 2, 7, 14, 18, 20, 22, 24 have insufficient quality (many of their features are illegible).
Line 213 - any references to the Tayfun model?
Line 246 - a reference to Cox and Munk?
(The are also minor punctuation/writing style issues and multiple colours of the text, but they are not crucial in understanding the paper).
Reviewer 2 Report
The comment of remotesensing-2173457 is given as following: This research mainly analyzed the effects of wind wave spectra, non-Gaussianity of sea surface, and swell on the distribution of facet normalized radar cross section (NRCS) simulated by the facet two-scale model (TSM) by comparing the simulated results with the Sentinel-1 SAR data, the Advanced Scatterometer (ASCAT) data, and the geophysical model function (GMF) at the wind speed range of 3-16 m/s, the wind direction range of 0°-360° , and the incidence angle range of 30°-50° under VV and HH polarizations. The topic seems to be relevant in the field, but no original method is proposed. Most previous studies of facet scattering were carried out based on the Gaussian sea surface constructed by E spectrum or JONSWAP spectrum. This research analyzed the effects of different wind wave spectra and non-Gaussianity of sea surface on the average scattering and scattering distribution under different incidence angles, wind speeds, and wind directions. This research result may be helpful when selecting spectrum under the simulation based on the TSM model. However, TSM is also a relatively old simulation metod with higher degree of approximation. The wave spectrum selection conclusion of this paper may not appropriate for other more precise simulation method. When comparing simulation data to Sentinel-1 data, it is better to use buoy data to obtain the information of sea surface winds and ocean waves in the corresponding area rather than ERA5 data. Buoy data may be more accurate. The circles marked with wavelengths shown in Figure 2 can also be added to Figure 18,Figure 20 and Figure 22 to visualize more information about the spectra. It is better to mark the physical meaning and unit of the color bar in the figures that involve color bars (Figure 2、17、18、19、20、21、22、24、27).
Reviewer 3 Report
This study investigated the effects of wind wave spectrum, non-Gaussianity, and swell on the prediction of NRCS with facet TSM. The TSM simulations are compared to NRCSs from SAR, ASCAT, and GMF. In general, this manuscript is well-written and presented. The figures are clear, as well as the discussions. I recommend this manuscript can be published as it is.
Some minor comments:
1. L112, ASCAT measurements contaminated by rainfall are rejected. how did you do that, since ASCAT can not provide rain information directly?
2. L116, when matching ASCAT and buoy data, the criteria shows a BUG. Within 25 km, one buoy data can match several ASCAT sigma0 data. I guess you only want to use the one with the closest distance.
3. L130, the latest C-band GMF is CMOD7, which is a refined version based on CMOD5.N.
Reviewer 4 Report
In this paper, the effects of wind wave spectra, non-Gaussianity of sea surface, and swell on the distribution of facet normalized radar cross section (NRCS) simulated by the facet two-scale model (TSM) are analyzed by comparing the simulated results with the Sentinel-1 SAR data, the Advanced Scatterometer (ASCAT) data, and the geophysical model function (GMF) . The experiments are performed. Overall, the experiments are detailed and solid. I have only some minor suggestions.
1. In line 5, JieZhang should be Jie Zhang.
2. In line 17, archieves -> achieve?
3. In line 128, “φ is the azimuthal wind direction angle” ->“and φ is the azimuthal wind direction angle”
4. In Table 2 and Table 3, dominant-> Dominant?
Round 2
Reviewer 1 Report
Thank you very much for the modifications, the paper is now ready!